



# Development of a forecast-oriented km-resolution ocean-atmosphere coupled system for Western Europe and evaluation for a severe weather situation

Joris Pianezze[1,a], Jonathan Beuvier[1], Cindy Lebeaupin Brossier[2], Guillaume Samson[1], Ghislain Faure[2], and Gilles Garric[1]

[1]Mercator Ocean International, Toulouse, France
[2]CNRM, Université de Toulouse, Météo-France, CNRS, Toulouse, France
[a]*now at: Laboratoire d'Aérologie/OMP, Université de Toulouse, CNRS, UPS, UMR5560, Toulouse, France*

**Correspondence:** *J. Beuvier (jonathan.beuvier@mercator-ocean.fr)*

**Abstract.** To improve high-resolution numerical environmental prediction, it is essential to represent ocean-atmosphere interactions properly, which is not the case in current operational regional forecasting systems used in Western Europe. The objective of this paper is to present a new forecast-oriented coupled ocean-atmosphere system and its evaluation. This system uses the state-of-the-art numerical models AROME (cy43t2) and NEMO (v3.6) with a horizontal resolution of 2.5 km. The

5  OASIS coupler (OASIS3MCT-4.0), implemented in the SurfEX surface scheme and in NEMO, is used to perform the communications between models. The evaluation of this system is carried out using 7-day simulations from 12 to 19 October 2018, characterised by extreme weather events (storms and heavy precipitation event) in the area of interest. Comparisons with in-situ and L3 satellite observations show that the fully coupled simulation reproduces quantitatively well the spatial and temporal evolution of the sea surface temperature and 10 m wind speed. Sensitivity analysis to OA coupling show that the use of an

10  interactive and high resolution SST, in contrast to actual NWP where SST is persistent and at low resolution, modifies the atmospheric circulation and the location of heavy precipitation. When compared to the operational-like ocean forecast, simulated oceanic fields show a large sensitivity to coupling. Forced ocean simulations highlight that this sensitivity is mainly controlled by the change in the atmospheric model used to drive NEMO (AROME vs. ECMWF IFS operational forecast). The oceanic boundary layer depths can vary by more than 40%. This impact is amplified by the interactive coupling and is attributed to

15  positive feedback between sea surface cooling and evaporation.

## Contents







# 1 Introduction

Ocean-atmosphere feedbacks occur over a wide range of spatial and temporal scales. They play a critical role in the evolution of climate (Intergovernmental Panel on Climate Change, 2014) but also in the evolution of smaller spatial and temporal

scales phenomena like tropical cyclones (Bender and Ginis, 2000; Smith et al., 2009; Jullien et al., 2014), mid-latitudes storms (Mogensen et al., 2018; Bouin and Lebeaupin Brossier, 2020b), sometimes leading to heavy precipitation events as for instance in the Mediterranean region (Rainaud et al., 2017; Meroni et al., 2018), dense water formation (Carniel et al., 2016; Lebeaupin Brossier et al., 2017), and ocean dynamics in particular in response to strong wind (e.g. Pullen et al., 2006; Small et al., 2012; Renault et al., 2019b; Jullien et al., 2020). It is therefore essential to represent them in numerical models to

correctly predict atmosphere and ocean dynamics for climate, environmental or weather applications.

Since the 1960s, global coupled ocean-atmosphere systems are indeed developed and used to investigate the future climate change (e.g. Meehl, 1990; Eyring et al., 2016) and, later on, served for seasonal forecasts (e.g. Stockdale et al., 1998). With the increase of High Performance Computer (HPC) resources (Shukla et al., 2010), many regional coupled research systems have been developed since the 2000s' (e.g. Bao et al., 2000; Chen et al., 2010; Warner et al., 2010; Voldoire et al., 2017) and

it is now possible to reach coupled ocean-atmosphere simulation on dedicated regions with an horizontal resolution of only few kilometers for both components (e.g. Small et al., 2011; Grifoll et al., 2016; Rainaud et al., 2017; Pianezze et al., 2018; Lewis et al., 2019). At that resolution, (i) atmospheric model represents explicitly the deep convection, the major gravity waves and the main interactions with orography (Weusthoff et al., 2010) and (ii) oceanic model is classified as eddy-rich resolution solving major baroclinic oceanic eddies (Hewitt et al., 2020).

Among these new kilometric ocean-atmosphere coupled systems, only few aim to operational oceanography purposes or Numerical Weather Prediction (NWP) applications, and even less are run operationally despite spread motivations and common interests (Brassington et al., 2015; Pullen et al., 2017). The main obstacles to this remain in particular the computing costs of an atmospheric model for operational oceanography, and, in general, a lower expertise on one or the other of the components and the absence of coupled initialisation strategy and dedicated validation tools.

To step forward, Météo-France and Mercator Ocean International (MOI) recently join their development efforts to build a new forecast-oriented coupled system based on two models used for operational purposes, which is presented in this paper. This new coupled system is an extension and update of the ocean-atmosphere coupled system developed by Rainaud et al. (2017) and Lebeaupin Brossier et al. (2017), that involves the regional non-hydrostatic NWP system of Météo-France, AROME, and, NEMO, the ocean model operated routinely by MOI for ocean forecasting. This new configuration covers Western Europe

and the western part of North-Africa and includes the Western Mediterranean Sea (up to Sicily eastwards) and also part of the North-East Atlantic Ocean, the English Channel and the North and Irish Seas (Fig. 1). This region is characterised by fine-scale ocean structures: estuaries and regions of freshwater influence related to large river plums (e.g. Simpson et al., 1993; Brenon and Le Hir, 1999; Estournel et al., 2001; Bergeron, 2004); thermal fronts notably in the French Atlantic continental shelf area (Yelekçi et al., 2017) and in particular the Ushant front of tidal origin (Chevallier et al., 2014; Redelsperger et al.,

2019), or also, the North Balearic Front in the Western Mediterranean Sea (García et al., 1994); slope current, wind-driven



circulation and mesoscale eddies in the Bay of Biscay (van Aken, 2002; Le Boyer et al., 2013); gyres in the Alboran Sea (Viúdez et al., 1998); meanders of the Algerian Current and eddies (Millot et al., 1990; Millot and Taupier-Letage, 2005); shelf circulation, cyclonic gyre, ocean deep convective area and Northern Current in the Gulf of Lions (e.g. Millot, 1991; Echevin et al., 2003; Testor et al., 2018; Carret et al., 2019). Furthermore, it is also frequently affected by several kinds of natural hazards of weather origin: strong wind related to storm, cyclogenesis (Trigo et al., 2002; Trigo, 2006) with for some cases an explosive development (Liberato et al., 2013) or even tropical-like characteristics (namely medicanes, Miglietta and Rotunno, 2019), sometimes interacting locally with the coast and/or orography (like mistral and tramontane, Bastin et al., 2006; Obermann et al., 2018); thunderstorms (Taszarek et al., 2019) including Mediterranean heavy precipitation events with floods (Ducrocq et al., 2016); heat waves (De Bono et al., 2004; Darmaraki et al., 2019; Ma et al., 2020); on which ocean-atmosphere interactions play a significant role. Better representing the air-sea feedback that occurs at fine-scale in this area is therefore relevant and developing a dedicated ocean-atmosphere coupled prediction system appears now essential to improve the high-resolution regional forecasts on both sides.

In that way, our common scientific objectives in this development between Météo-France and MOI are (1) to share and improve knowledge about fine-scale ocean-atmosphere interactions in this wider region, (2) to be able to provide high-resolution and consistent atmosphere and ocean forecasts over Western Europe and notably the entire French coastal area, including the Corsican coasts, and (3) to prepare a coupled initialisation strategy also able to ensure the consistency with the large-scale driver models used at the boundaries.

The new coupled system and the coupling strategy are presented in Section 2. Sections 3 and 4 present the experimental design and the evaluation of this new coupled system, as the coupling impacts for both atmospheric and oceanic forecasts. Finally, conclusions and perspectives are given in Section 5.

## 2 Description of the new coupled system

In this section the models and the coupling strategy used in this new coupled system are presented.

### 2.1 Atmospheric and surface models

The atmospheric model used in this new coupled system is the cycle 43 (cy43t2) of the non-hydrostatic Application de la Recherche à l'Opérationnel à Méso-Échelle (AROME) NWP regional model (Seity et al., 2011; Brousseau et al., 2016). The AROME physical configuration used here is close to the one operationally used at Météo-France but covers a wider area [than the AROME-France NWP 1.3 km-resolution model] around Western Europe (Fig. 1), with a 2.5 km-resolution and is run here without data assimilation.

In more details, AROME has $1285 \times 1789$ horizontal grid points and a vertical grid of 90 hybrid $\eta$-levels with a first-level thickness of almost 5 m. The advection scheme in AROME is semi-Lagrangian and the temporal scheme is semi-implicit with a time-step of 50 s. The 1.5-order turbulent kinetic energy scheme from Cuxart et al. (2000) is used. For the purpose of this study, the Current-FeedBack effect (CFB) has been added in the turbulent scheme of AROME, following Renault et al.

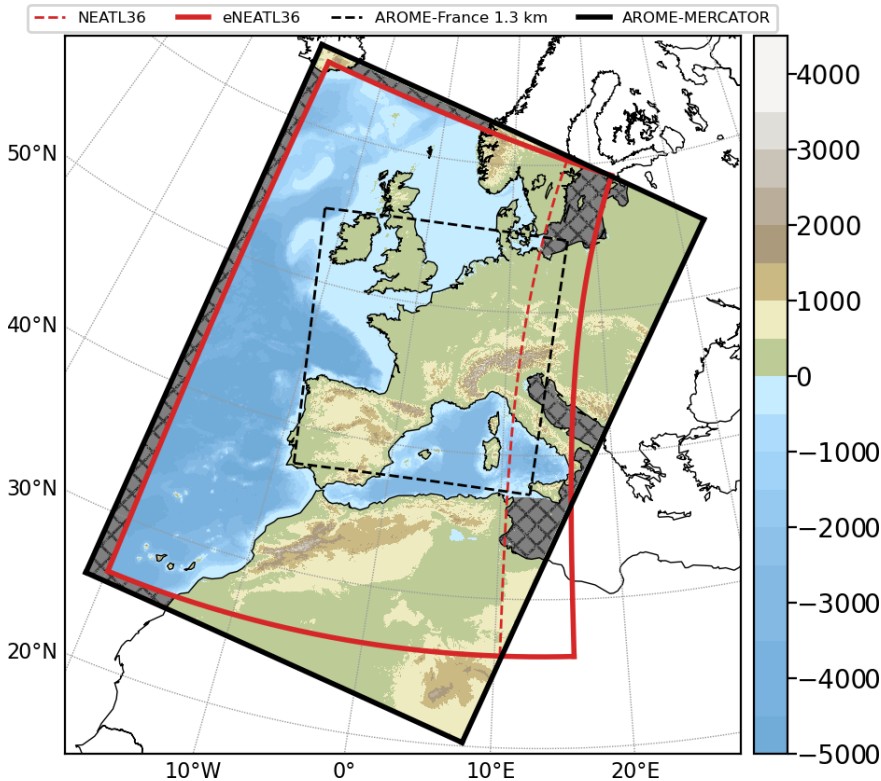

**Figure 1.** Simulation domain illustrated by the bathymetry [m] in NEMO (in blue) and by the orography [m] of the AROME model (in green-brown colors). The lines indicate the boundaries of NEMO-eNEATL36 configuration (red) and of the AROME-Mercator domain (black). For AROME-Mercator, the grey marine zones are always uncoupled (constant initial SST and null current are used, see text). The dashed lines indicate the boundaries of the actual operational configurations of AROME (AROME-France, 1.3 km-resolution in black) and NEMO over the Iberia-Biscay-Ireland (IBI) region (NEATL36, 1/36°-resolution in red).

(2019a) and based on the exact same developments as previously done in the MESO-NH model (Bouin and Lebeaupin Brossier, 2020a). Thanks to its 2.5 km horizontal resolution the deep convection is explicitly resolved while the shallow convection is
parameterized with the Eddy Diffusion Kain Fritsch EDKF, (EDKF, Kain and Fritsch, 1990) scheme. The ICE3 one-moment microphysical scheme of Pinty and Jabouille (1998) is used to compute the evolution of five hydrometeor species (rain, snow, graupel, cloud ice and cloud liquid water). Radiative transfer is based on Fouquart and Bonnel (1980) scheme for short-wave radiation and the Rapid Radiative Transfer Model (RRTM, Mlawer et al., 1997) for long-wave radiation.

The surface exchanges are computed by the SURFace EXternalisé (SURFEX) surface model (Masson et al., 2013) consider-
ing four different surface types: land, towns, sea and inland waters (lakes and rivers). Output fluxes are weight-averaged inside each grid box according to the fraction of each respective tile, before being provided to the atmospheric model at every time step. Exchanges over land are computed using the ISBA (Interactions between Soil, Biosphere and Atmosphere) parametrization (Noilhan and Planton, 1989). The formulation from Charnock (1955) is used for inland waters, whereas the Town Energy





Balance (TEB) scheme is activated over urban surfaces (Masson, 2000). For the sea surface, the albedo is computed following

the Taylor et al. (1996) scheme and sea surface fluxes are computed with COARE3.0 parametrization (Fairall et al., 2003).

Like when run operationally, AROME in this configuration can be initialised and forced at its lateral boundaries by operational global analyses and/or forecasts from Action de Recherche Petite Echelle Grande Echelle (ARPEGE ; Courtier et al. (1991)) or Integrated Forecasting System (IFS ; ECMWF (2020). No lateral condition is applied on SurfEx which is initialized over continental surfaces with the ARPEGE surface analysis.

## 2.2   Oceanic model

The oceanic model used in this coupled system is based on the version 3.6 of the Nucleus for European Modelling of the Ocean model (NEMO, Madec et al., 2017). It is a state-of-the-art primitive-equation, split-explicit, free-surface oceanic model. It has been built from the operational IBI configuration (Maraldi et al., 2013; Sotillo et al., 2015; Gutcknecht et al., 2019; Sotillo et al., 2021), spatially extended in the Mediterranean Sea (see Figure 1). The meridonal boundary in the IBI operational

configuration between the Gulf of Genoa, Corsica, Sardinia and Tunisia, has been moved to a zonal boundary between Tunisia and Sicily; thus this new regional configuration now covers the entire Tyrrhenian Sea. The horizontal resolution is 1/36° with $1294 \times 1894$ horizontal grid points and the vertical grid contains 50 stretched z-levels. The vertical level thickness is 0.5 m at surface and around 450 m for the last levels (i.e. at 5700 m depth).

Temporal scheme for both tracer and momentum is a leapfrog scheme associated to Robert-Asselin filter to prevent model

instabilities (Leclair and Madec, 2009). The free surface is explicit with time splitting and a baroclinic time step of 150 s and a barotropic time step 50 times smaller. Momentum advection is computed based on the vector invariant form while the Total Variation Diminishing (TVD) scheme is used for tracer advection in order to conserve energy and enstrophy (Barnier et al., 2006). The Generic Length Scale (GLS) scheme is used in that configuration which is based on  two prognostic equations: one for the turbulent kinetic energy, and another for the generic length scale (Umlauf and Burchard, 2003, 2005). Open boundaries

conditions (OBC) are based on the 2D characteristic method (Blayo and Debreu, 2005). The atmospheric pressure component is added hypothesizing pure isostatic response at open boundaries (inverse barometer approximation). Rivers freshwater inputs are imposed part as OBC in the domain locations for 33 main rivers and part as a coastal runoff to close the water budget from land. The tidal forcing is prescribed from the FES2014 dataset (Carrere et al., 2015) and applied as unstructured boundary in the NEMO domain.; 11 tidal harmonics (M2, S2, N2, K1, O1, Q1, M4, K2, P1, Mf, Mm) are used. Solar penetration is

parameterized according to a five-bands exponential scheme (considering the UV radiations) function of surface chlorophyll concentrations, using a monthly climatological version of the CMEMS ESA-CCI product covering the North East Atlantic area (OCEANCOLOUR_ATL_CHL_L4_REP_OBSERVATIONS_009_091, Colella et al., 2020).

In that new configuration, version 2.0 of the eXtensible Markup Language XML Input/Output Server (XIOS, Meurdesoif, 2013) is used to manage NEMO output files.

The model is initialised by fields from the operational IBI configuration at 1/36° (IBI36, Sotillo et al., 2021) on the common domain (see Figure 1) and from the global CMEMS configuration at 1/12° (GLO12, Lellouche et al., 2018) in the Tyrrhenian Sea, and forced at the OBC with daily analyses from this CMEMS GLO12 configuration.





## 2.3 Coupling strategy

Communications between AROME/SurfEx and NEMO models are performed with the Ocean-Atmosphere-Sea-Ice-Soil cou-
pler (OASIS3-MCT_4.0, Valcke (2013); Craig et al. (2017)). OASIS3-MCT is a library allowing synchronised exchanges of
coupling information between different numerical models. OASIS calls were inserted in SurfEx sources by Voldoire et al.
(2017) allowing the atmosphere-ocean coupling between AROME/SurfEx and NEMO.

A similar coupling algorithm as Rainaud et al. (2017) and Lebeaupin Brossier et al. (2017) is used in this study and is only
summarised here and in Table 1 for clarity. AROME-SurfEx sends to NEMO the net non-solar heat flux, the two components
of the wind stress and the net freshwater flux computed for the sea tile only, which are then imposed at the surface boundary
condition of NEMO. The solar heat flux is also send to NEMO and is used to calculate the penetrative radiation in the ocean.
Contrary to Rainaud et al. (2017), Lebeaupin Brossier et al. (2017), but also Arnold et al. (2020), the possibility of exchanging
atmospheric surface pressure was implemented in this study and is also exchanged interactively during the coupled simulations
for the inverse barometer approximation. In return, NEMO sends to AROME-SurfEx, the sea surface temperature and the sea
surface current components that then enter in the sea surface turbulent fluxes computation and in the atmospheric turbulence
scheme.

The remapping files needed to interpolate fields between NEMO and AROME-SurfEx with a distance weighted nearest-
neighbour interpolation method using four neighbours are created offline using OASIS tools. Where the ocean is masked
because outside the NEMO domain (hashed area in Fig. 1), AROME uses a SST constant in time and equal to the one used at
the initial time, and the surface currents taken are always equal to zero (see also Appendix A1).

**Table 1.** Variables exchanged between NEMO (O) and AROME/SurfEx (A) via the OASIS3-MCT coupler.

| Variable | Description | | Units |
|---|---|---|---|
| $Q_{ns}$ | Non solar heat flux | A → O | $W.m^{-2}$ |
| $Q_{sr}$ | Solar heat flux | A → O | $W.m^{-2}$ |
| $\tau_{x,y}$ | Momentum flux | A → O | $N.m^{-2}$ |
| E-P | Evaporation minus precipitation | A → O | $kg.m^{-2}.s^{-1}$ |
| $P_{atm}$ | Atmospheric surface pressure | A → O | Pa |
| SST | Sea surface temperature | O → A | K |
| $u_{cur}$, $v_{cur}$ | Sea surface currents | O → A | $m.s^{-1}$ |




## 3 Numerical set-up

### 3.1 Case study : storms and high precipitation (12-19 October 2018)

The evaluation of this coupled system is carried out through 7-day simulations of a case study from 12 to 19 October 2018. During these seven days, Western Europe experienced a severe weather sequence (see Fig. 2) with a mid-latitude storm (Callum), two [ex-]tropical cyclones (Leslie and Michael) and a Mediterranean heavy precipitating event (Aude HPE case).

**Figure 2.** Illustrations of the case study: (a) True color image of Terra/MODIS (source: https://worldview.earthdata.nasa.gov/) on 11 October 2018 over the North Atlantic Ocean showing the storm Callum and the Leslie and Michael hurricanes (arrows depict their trajectories towards the area of interest); (b) Rainfall totals (mm) from 11 to 12 October 2018 over Wales (Callum's impacts, Figure 64 from Kendon et al., 2019, source: MetOffice); (c) Wind gust observations (km/h) over Iberian Peninsula on 13 October 2018 around 23 UTC (Leslie's landfall, source: www.meteociel.fr); (d) Rainfall amounts (mm) between 06 UTC on 14 October and 06 UTC on 15 October 2018 over the French Languedoc region (Aude event, source: Météo-France - edited 19/02/2019).




In more details, storm Callum was named by Met Éireann on 10 October when it was forecast to affect the British Islands and more particularly Ireland and Wales. The storm deepened over the Atlantic Ocean on 11 October, reaching a minimum pressure depth of 938 hPa. On 12 October, strong wind affected Ireland and the north-western Wales, with gust up to 140 km/h at Capet Curig. Heavy rainfall also occurred over Wales (Fig. 2b), in particular inland due to an orographic enhancement, with
up to 219 mm in 36 hours recorded at Libanus (Powys) making Callum one of the most severe rainfall events across Wales in the last 50 years (Kendon et al., 2019). Storm Callum had indeed strong impacts due to flooding, also because the wind peak coincided with high spring tides and led to large waves, with some coastal flooding, largely enhanced by the heavy rainfall.

Hurricane Leslie was a large, long-lived, and very erratic tropical cyclone over Atlantic. Followed by the National Hurricane Center (NHC) since 23 September (Pasch and Roberts, 2019), it stroke the Iberian Peninsula on the evening of 13 October.
For the first time on record, a Tropical Storm Warning was issued for Madeira Island. In fact, after a stationary position in the Eastern Atlantic at the beginning of October, Leslie started moving and intensifying under favourable environment with slightly warmer water, so it re-attained the hurricane status on 10 October. Leslie reached its peak intensity with maximum sustained winds of 150 km/h and a minimum central pressure of 968 hPa, on 00 UTC 12 October, about 1000 km south-southwest of the Azores. While then re-weakening, Leslie raced east-northeastwards, accelerated by the mid-latitudes westerlies, and passed
about 320 km North-Northwest of the Madeira Island on 06 UTC, 13 October. At 18 UTC, Leslie became a strong extratropical cyclone, at about 190 km West-Northwest of Lisbon. Leslie's extratropical remnant made finally landfall close to Figueira da Foz (Coimbra District) just after 21 UTC with wind gusts above 110 km/h (Fig. 2c), heavy rains and strong waves. Spain was also affected by strong wind with up to 96 km/h in Zamora (Castile and Leòn). Leslie cyclone's centre became ill-defined after it moved over the Bay of Biscay on 14 October. At the same time, it induced favourable and steady conditions for heavy rainfall
in the Western Mediterranean, Leslie remnant acting as a large trough generating a southerly flow.

As described in Caumont et al. (2021) and Mandement and Caumont (2020), in the night of 14 to 15 October 2018 the Languedoc region in the south of France, was indeed affected by heavy rainfall caused by a regenerative multi-cellular convective system organised along a convergence line between the moist southerly low-level flow and a quasi-stationary cold front over south-western France along a mean sea level pressure (MSLP) trough that linked Leslie to a low located over Ireland over
south-western France. During the evening and night of 14 to 15 October, a low rapidly deepened around the cold front and induced a strong convective activity over the Catalan Sea, between the Balearic Islands and Valencia region. The most intense rainfall occurred between 19 UTC 14 October and 07 UTC 15 October. The Météo-France quantitative precipitation estimation gives a maximum 24 h-accumulated rainfall total of 342 mm close to Trèbes (Aude, Fig. 2d). Intense rainfall mainly occurred in less than 12 hours, leading to flash floods in particular in Villegailhenc (Aude), and caused 15 fatalities.
Some days sooner, the extratropical cyclone Michael emerged into the Atlantic around 06 UTC 12 October after passing near Norfolk (Virginia, US). Michael re-obtained hurricane-force winds on 13 October in the Atlantic waters south of Nova Scotia and Newfoundland, then quickly travelled within westerlies to the North-Eastern Atlantic on 14 October. The cyclone turned sharply southeastward and later southward around the northeastern edge of the subtropical ridge, weakening slightly, as it approached the Iberian Peninsula. Michael dissipated by 00 UTC on 16 October, while it was located just west of northern
Portugal and just after Leslie's remnant was absorbed into Michael's remnant, following a brief Fujiwhara (1921) interaction.





This 7-day period was chosen as the weather situation encountered is known to foster large air-sea interactions, but also because both ocean and weather forecasts may exhibit a larger sensitivity to coupling in such conditions. This is analysed through different simulations in the coupled and forced modes that are described in the following Section.

## 3.2 Experiments

To evaluate the ocean-atmosphere coupling impact on the atmospheric and oceanic forecasts, four experiments were performed and are detailed below and in Table 2.

The OA experiment is the ocean-atmosphere coupled forecast over 7 days, starting on 12 October 2018 00 UTC. The initial atmospheric conditions comes from the global IFS analysis of 12 October 2018 00 UTC and the lateral atmospheric forcing comes every 6 hours from the global IFS forecast starting on 12 October 2018 00 UTC. The ocean initial fields come from the

combination, as described in 2.2, of the CMEMS IBI and GLO analyses (3D daily fields of the 11 October) and OBC for the 7 days come from the CMEMS GLO daily analyses. The ocean-atmosphere coupling period is set to 600 s, *i.e.* the fields are exchanged every 4 NEMO time-steps and 12 AROME time-steps.

The reference experiment for atmospheric forecast (ARO) is similar to the OA experiment except that, as uncoupled, (i) the SST is kept persistent in time and (ii) sea surface currents are not taken into account. Note that this ARO experiment is

equivalent to one operational deterministic execution of AROME at Météo-France (called *AROME-IFS*), but with two adaptations. First, the lateral atmospheric conditions frequency is changed to 6 hrs in order to be able to run over a 7-day period (against 42 to 48h for AROME operational forecasts). This was mandatory due to less frequent forecast outputs available for the longest-term ranges of IFS. And secondly, for consistency with OA, the initial SST field is the combination of the PSY4 and IBI36 SST fields (instead of the ARPEGE SST analysis for AROME-IFS). Thus, comparing ARO with OA allows to evaluate

the ocean-atmosphere coupling impact, *i.e.* the effect of an interactive evolution of SST and the impact of taking currents into account, on the weather forecast.

Two ocean-only experiments were also run. OCE-ifs is the simulation close to the operational mode of IBI36, *i.e.* the initial conditions consist in the combination of the CMEMS IBI and GLO analyses (3D daily fields of the 11 October) and OBC for the 7 days come from the CMEMS GLO daily analyses (similarly to the ocean component of OA). The atmospheric forcing

uses the bulk variables from IFS (2 m-air temperature, 2 m-humidity, 10 m-wind components, rainfall, mean sea level pressure, short-wave and long-wave solar fluxes) and the IFS bulk parametrization available in the NEMO surface scheme (meaning the SST evolution and sea surface currents are taken into account to compute the air-sea exchanges). OCE-aro is an intermediate simulation using the ARO (AROME) bulk variables as atmospheric forcing (the same bulk variables as for IFS are used except for the wind speed which is taken at 5 m, the height of first vertical level of AROME) and the COARE3.0 sea surface turbulent

flux parametrization (Fairall et al., 2003) through SURFEX offline. Comparing OCE-aro with OA on one hand and OCE-aro with OCE-ifs on the other permits to disentangle the ocean-atmosphere coupling effect on the ocean forecast from the impact of the atmospheric forcing change.





**Table 2.** Set of simulations.

| Name of simulation | Type of simulation | Forcing/coupling time-step | Fluxes param. |
| --- | --- | --- | --- |
| OA | Fully coupled OA | 600 s | SFX-COARE3.0 |
| ARO | AROME forced by persistent SST equal at $SST^{ini}$ and no oceanic currents | - | SFX-COARE3.0 |
| OCE-ifs | NEMO forced by bulk variables from IFS | 3600 s | NEMO-IFS |
| OCE-aro | NEMO forced by bulk variables from ARO simulation | 3600 s | SFX-COARE3.0 |

## 4 Simulation results

This section presents an evaluation of the coupled OA simulation (Section 4.1), the respective impacts of the high-resolution
interactive atmosphere on the oceanic forecast (Section 4.2) and of the coupled ocean on the atmospheric forecast (Section
4.3). For comparison purposes, most of the figures presents the results of several simulations. Therefore, the reader may come
back to figures following the text.

### 4.1 Evaluation of the OA coupled simulation

This section describes the OA coupled simulation and presents its evaluation in comparison with observations available at the
sea surface and in the boundary layers.

#### 4.1.1 Sea surface temperature

At the initial state of OA (as for all the simulations), a latitudinal SST gradient is visible, from 7°C in the northwest to more than
24°C in the southwest part of the domain and in the Mediterranean sea (Fig. 3a). Small-scale structures in SST are also visible
and are related to the presence of mesoscale oceanic eddies, resolved at that 1/36° horizontal resolution (or partly resolved in
the Mediterranean part). After 1 (Fig. 3b) and 3 (Fig. 3c) simulated days, the signatures of Callum, Leslie and Mickael storms
are visible with an associated sea surface cooling of up to 2.5°C persisting during the 7 simulated days (Fig. 3d). This cooling
is mainly due to oceanic vertical mixing processes enhanced by the strong wind produced by these storms. At the end of the 7
simulated days, the average temperature over the domain is 0.6°C colder than initially with local differences varying up to 35%
of the initial SST (cooler or warmer depending of the location). The maximum differences are located in the areas of influence
of the storms (Atlantic ocean).

In Figure 4, the simulated sea surface temperature is compared to satellite observations coming from the Copernicus Ma-
rine Service portal (http://marine.copernicus.eu). This L3 SST is obtained from several satellite sensors which are combined
together and interpolated on a regular 0.02° grid, and is available every day with daily average. In order to be able to compare
the simulated and observed SST fields, it is necessary to interpolate the simulated SST on the satellite observation grid taking
into account the masked areas related to the presence of clouds and therefore where no satellite data is available (Fig. 4a and
4d). Whether at the beginning or at the end of the simulation, the simulated SST is close to the observed SST with a mean bias
of less than 0.1°C. Differences can be noted in the position of oceanic structures, which leads to local differences in SST that


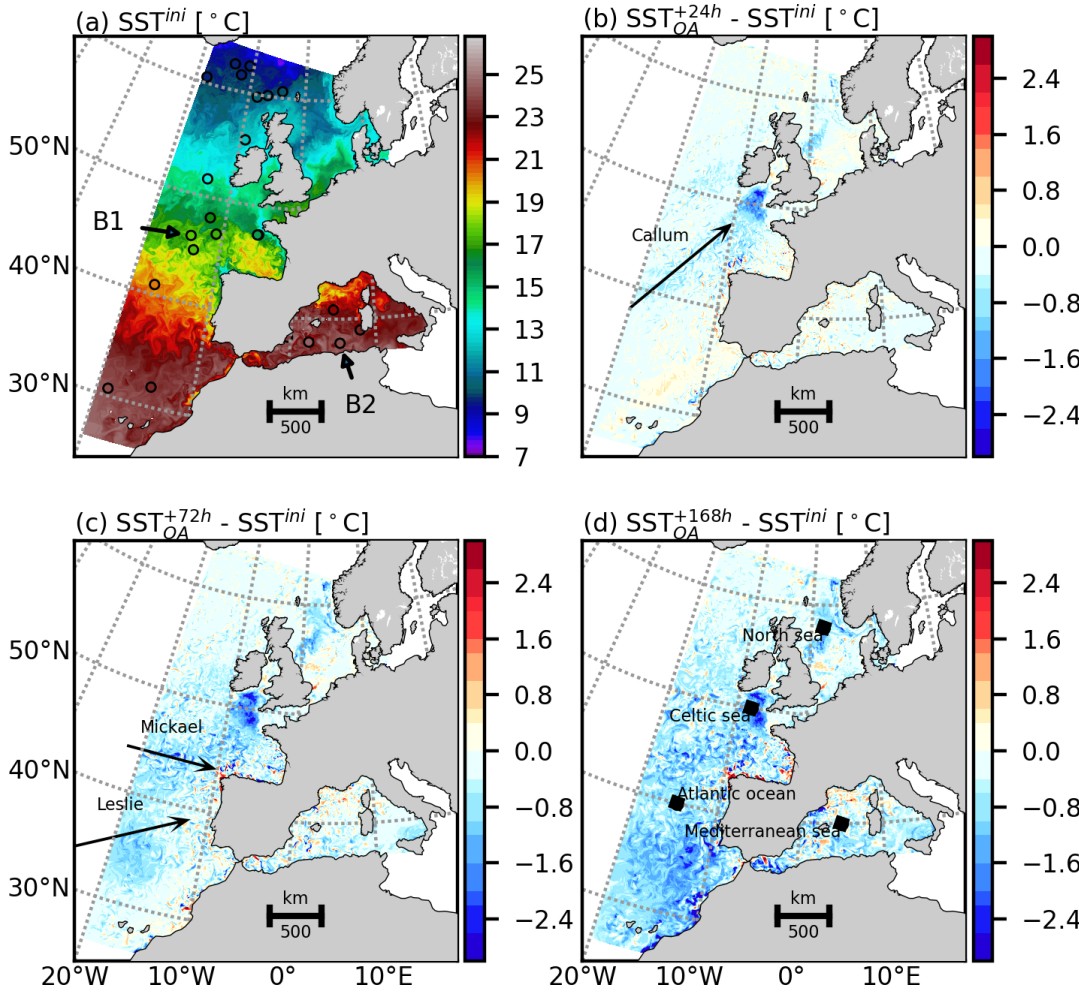

**Figure 3.** Initial [12 Oct. 2018 00 UTC] (a) and evolution of the SST (°C) after 1 day (b), 3 days (c) and 7 days (d) in the coupled simulation (OA; Table 2). In (a), the colour circles represent the SST measured by drifting buoys at that time ; B1 and B2 labels indicate the location of the two drifting buoys used in Figure 5. Black squares in (d) correspond to four extracted areas used for analyse in the next subsections.

can vary by up to ±4°C. In addition, it can be noted that the simulated cooling present in the Celtic Sea is stronger than the observed one.

Temporal evolution of simulated sea surface temperature is also compared to in-situ observations (drifting buoys) available on the Coriolis project portal (http://www.coriolis.eu.org) in Figure 5 (the locations of the observations used for the evaluation are shown in Figure 3a). Among the full observational data-set, we select only data which have almost fully time series during the 7 simulated days (33 drifting buoys), and with a hourly period (see B1 and B2 examples in Fig. 5a,b). Despite this selection, the high density of drifting buoys observation allows to evaluate the simulated SST over the entire domain. For all the buoys

represented in Figure 3a, statistics of the OA experiment are computed and are summarised in the Taylor Diagram in Figure


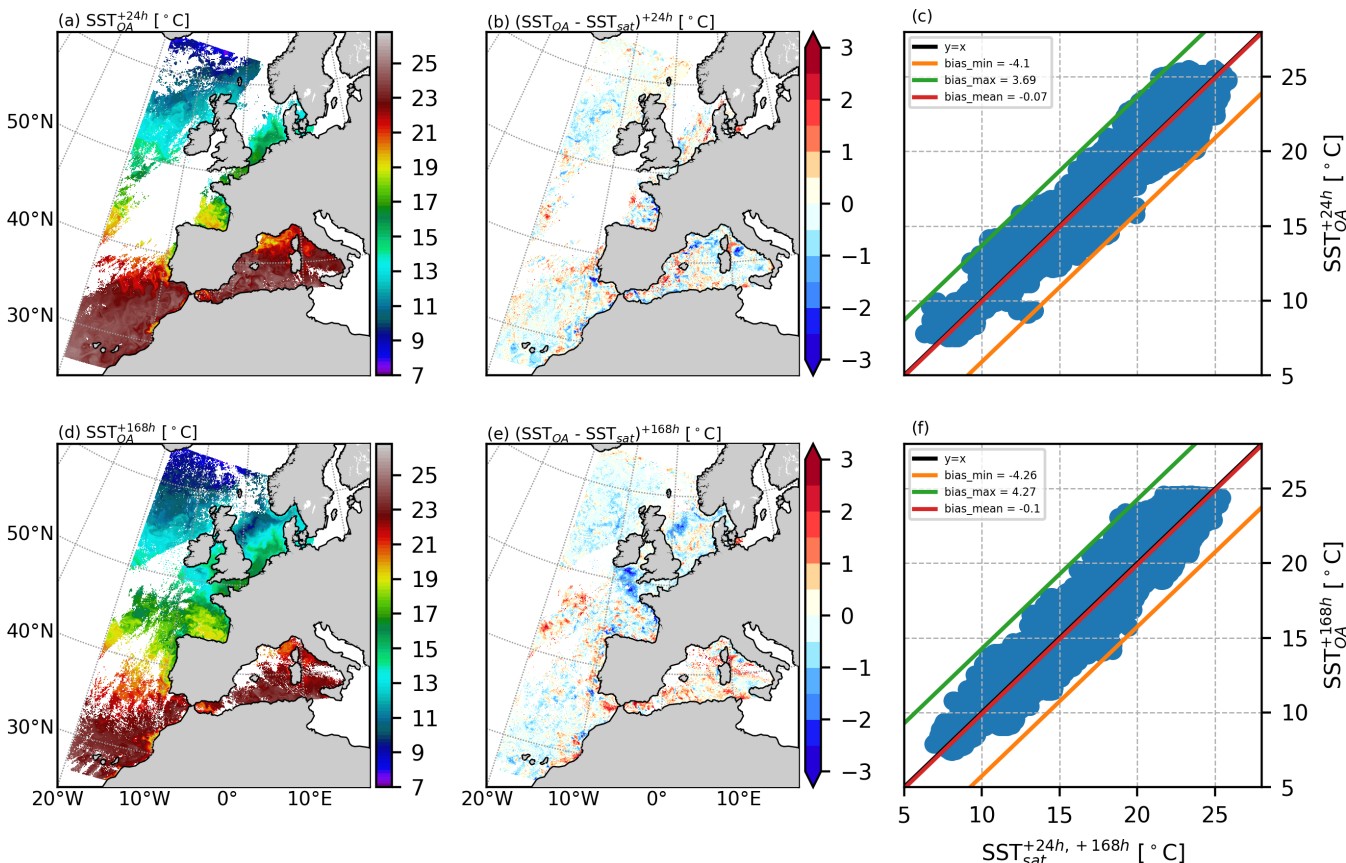

**Figure 4.** Comparison with SST satellite L3 observations: (a,d) OA simulated SST (°c) interpolated on the observed points, (b,e) differences between simulated and observed SST and (c,f) corresponding scatterplots; for 12 Oct. 2018 (a,b,c) and 18 Oct. 2018 (d,e,f).

5c. For OA, the mean bias is 0.04°C and the standard deviation is 0.2°C, but scores show a large variability. The correlation is 0.47 on average. The examples of B1 and B2 illustrate the good behaviour of OA in representing the weekly surface cooling. The rapid and intense SST variations are also reproduced, as visible for B1 (Fig. 5a), related to the storm Callum, or for the diurnal cycle seen at B2 (Fig. 5b), on 12 and 18 October for example in OA, however with differences in terms of intensity
with respect to observations.

In spite of local differences, the coupled simulation reproduces thus accurately the mean gradient, mesoscale structures and evolution of SST during the 7 simulated days.

In order to further evaluate the numerical experiments, we chose to focus on some dedicated locations, where intense air-sea
interactions are expected. For that, we define four boxes of 50 km × 50 km and their locations are visible in Figure 3d (black squares).





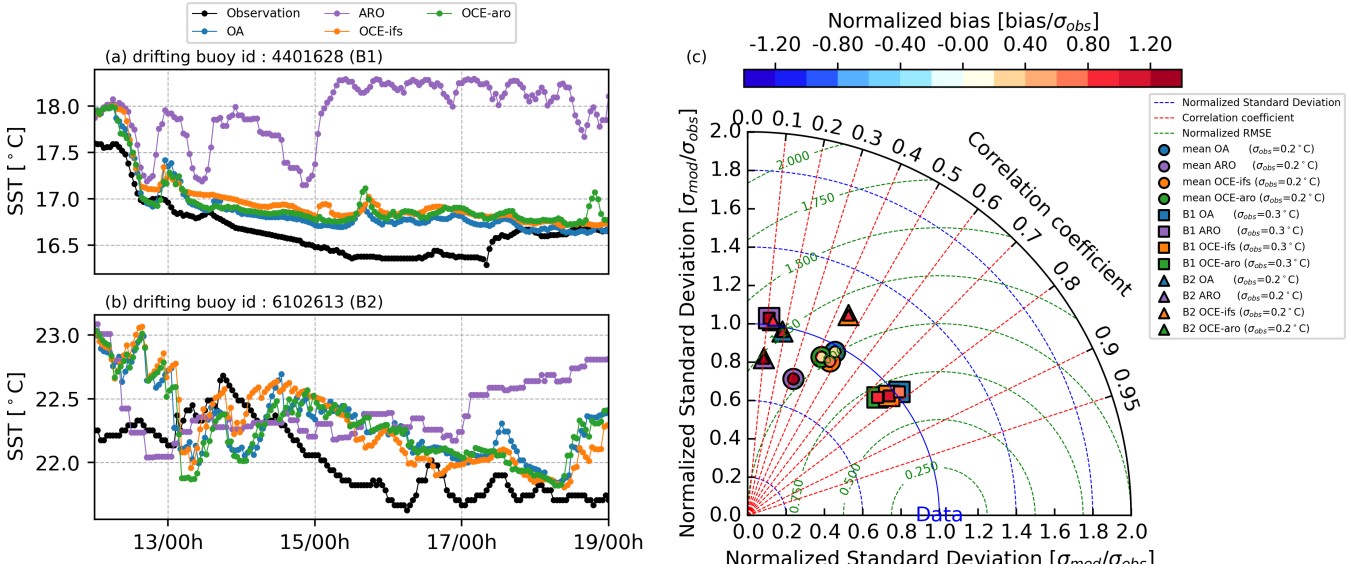

**Figure 5.** Temporal evolution of sea surface temperature observed and simulated at the location of the buoys B1 (a) and B2 (b). (c) Taylor diagram made from comparison with 33 selected buoys visible in Figure 3a. Mean statistics for the 33 selected buoys are represented in circles, statistics for buoy B1 only in squares and for buoy B2 in triangles. The inner colour indicates the normalised bias. The external colour indicates the experiment: blue for OA, purple for ARO, orange for OCE-ifs and green for OCE-aro.

Temporal evolution of sea surface temperature in these four boxes is presented in Figure 6a,b,c,d. As discussed in the previous paragraph, the simulated SST decreases during the 7 simulated days in OA, with diurnal variations visible in the Mediterranean sea at the beginning of the simulated period. In the Celtic and North seas, the sea surface temperature decreases

by more than 1.5°C and 0.5°C in less than 1 day, respectively. This intense sea surface cooling is attributed to the passage of Callum storm and its corresponding high wind speed ($> 20$ m s$^{-1}$, see Figure 6s and t).

#### 4.1.2 Sea surface dynamics, salinity and ocean mixed layer

As for the temporal evolution of sea surface temperature, the sea surface salinity (SSS), sea surface height (SSH) and sea surface currents (SSC) are extracted in the four locations (Fig. 3d, black squares) and are presented in Figure 6e to 6p.

In addition to SSS variations due to tide, the SSS time series show a global increase in the Mediterranean, Atlantic Ocean and North Sea (Fig. 6e,f,h). It reaches about $+0.04$ PSU day$^{-1}$ over the 7 simulated days in the Mediterranean and is twice lower for the two others (*i.e.* Atlantic Ocean and North Sea boxes). The strong evaporation fluxes linked to the presence of high winds are responsible for these increases (not shown). Only the Celtic Sea shows a decrease in SSS of $-0.15$ PSU in the first 36 simulated hours (Fig. 6g). This can be explained by the intense oceanic mixing associated to strong winds, which

tends to mix less salty water to the surface, while the precipitation associated with the passage of Callum does not contribute significantly to the decrease of SSS in this area (not shown).

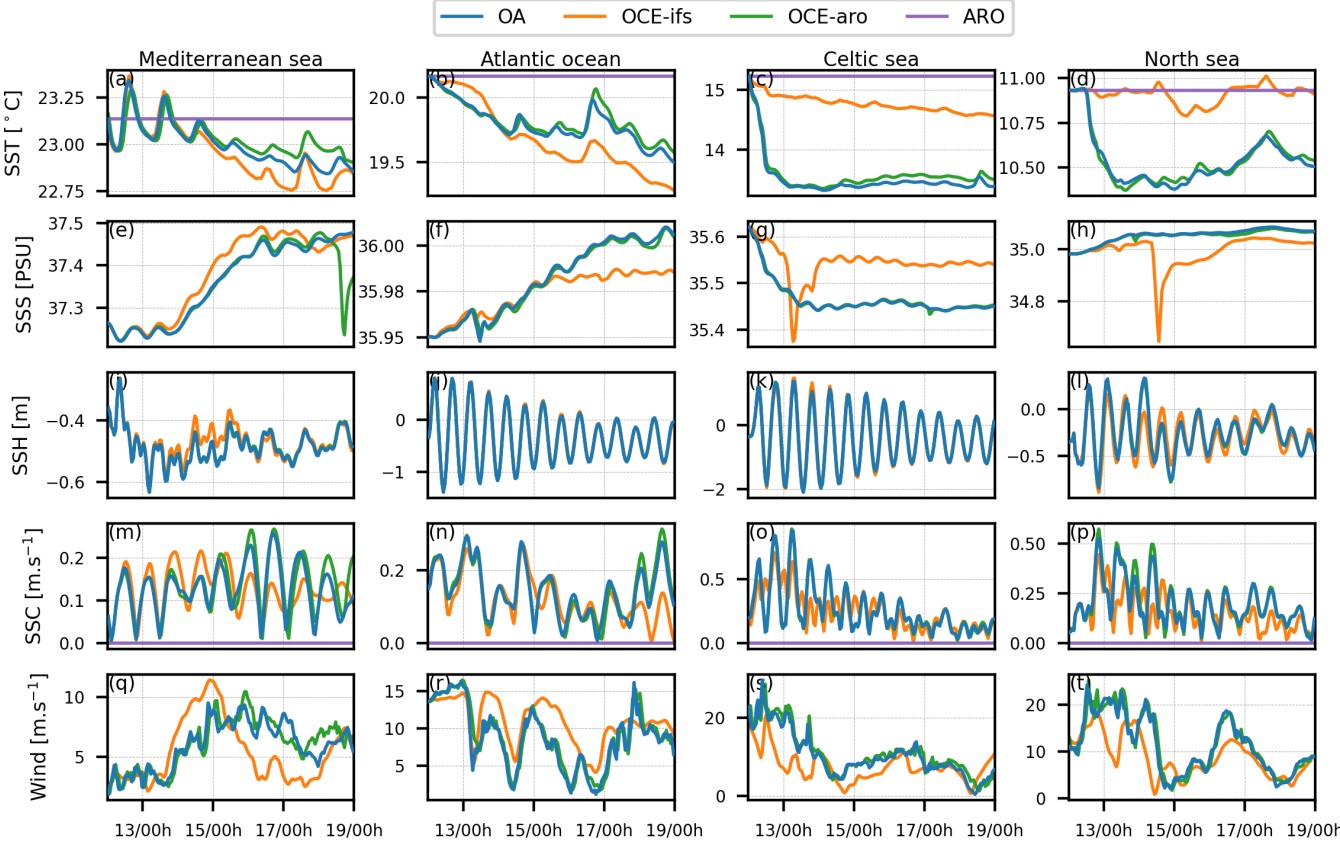

**Figure 6.** Temporal evolution of simulated sea surface temperature (SST, °C), salinity (SSS, psu), height (SSH, m) and current speed (SSC, m s$^{-1}$) and of near-surface wind velocity (m s$^{-1}$), extracted in the four areas presented in Figure 3d. In (q,r,s,t) ARO wind speed is the same as the OCE-aro one. Since the ARO simulation does not take into account the SSS and SSH, they are not represented in this Figure.

With respect to SSH variations (Fig. 6i,j,k,l), they are strongest in the Celtic Sea where the tidal amplitude is higher. The amplitude of these variations reaches 4 m and decreases over the 7 days, in relation to the decrease of the tidal coefficient from 95 on the $12^{th}$ to 30 on the $17^{th}$ (values for Brest harbour). In the Atlantic Ocean, the variation of SSH is also important

with an amplitude of one meter, while its weaker in the North sea, due to a smaller amplitude of the tidal harmonics in this area, leading also to a more variable signal related to interactions between these harmonics. In the Mediterranean sea, the SSH variations have the smallest amplitude ($\approx 0.2$ m), which are in fact mainly related to the presence of oceanic eddies.

In the coupled experiment (OA), the sea surface currents (SSC) are also exchanged. The spatial and temporal evolution of these currents are important during the 7 simulated days. Their intensity are maximum in the Channel, reaching more than 2

m.s$^{-1}$ locally, due to tidal currents (not shown). Temporal evolution of SSC in the four extracted areas are presented in Fig 6m,n,o,p. SSC are maximum in Celtic and North Seas, reaching more than 0.5 m s$^{-1}$ with intensities that vary with respect to the tides. For the Atlantic Ocean and Mediterranean Sea boxes, SSC intensity is less important but can reach up to 0.25 m s$^{-1}$.





The evolution of the ocean mixed layer is analysed more finely thanks to temporal evolution of temperature vertical profiles (Fig. 7). Black lines in Fig 7 correspond to ocean mixed layer depth (MLD). To compute this mixed layer depth, the potential

density field is used: for each grid point, the value at 10m depth is taken as a reference, and the mixed layer depth is obtained when the vertical difference is higher than 0.01 kg.m$^{-3}$ (pycnocline depth). At the beginning of the OA simulation, the MLD is around 40 m in the Atlantic Ocean, the Channel and the North Sea. In the Mediterranean, the MLD is thinner, around 20-30 m, corresponding to typical MLD values for late summer (D'Ortenzio et al., 2005). The MLD is stable in the Mediterranean and deepens slightly in the Atlantic, from 40 m to 50 m during the 7 days simulated. The strongest MLD variations are located

in the north-western part of the domain, in the Celtic Sea (Fig. 7g) and North Sea (Fig. 7j) boxes, where a significant deepening of the MLD is visible during the first simulated days. This MLD deepening reaches 35 m in the first simulated days in North sea and up to 65 m in Celtic sea. Callum storm and its associated high turbulent fluxes are responsible for this strong MLD deepening. After the passage of Callum, a slow restratification is simulated in the Celtic Sea from 14 October which is also present but less visible in North Sea.

On the last day of simulation (Fig. 8), the highest MLD values are found in the north-westernmost part of the domain, around 100 m deep, up to 150 m locally, and in the Celtic Sea (80-100 m). The smallest values (<30 m) are found in the coastal areas (in relation with lower SSS values in the river plumes) and in the Mediterranean Sea.

### 4.1.3 Wind

The OA simulated wind field is examined in Figure 9a,c,e and compared to in-situ wind measurements available in the Coriolis

database. It is important to note that the wind observations are set at a height of 10 meters, thus we use a 10-m diagnostic wind from AROME and not the pronostic 5-m wind values.

During the first simulated day (12 Oct., Fig. 9a), the storm Callum moves towards the British Islands, inducing strong wind (above 20 m.s$^{-1}$) over a wide area affecting Portugal to United Kingdom. Locally, wind speed value reaches the maximum value of 41.5 m s$^{-1}$ in the Celtic Sea. The comparison with data (circles in Fig. 9a) shows that OA overestimates wind speed at

that time. Regarding the M1 moored buoy (58.3°N-0.1°E, north-east of the coasts of Scotland), however, OA reproduces quite well the first wind peak in the afternoon of 12 Oct., but simulates a too strong and too early second peak on 13 Oct (Fig. 10a).

On 15 Oct. 00UTC (Fig. 9b), OA simulates a wind structure related to the remnants of Michael and Leslie close to Galicia. The comparison to buoy observations shows a good correspondence, even if wind measurements are mainly localised close to the coasts and miss the stronger wind area. Moderate wind (13 m s$^{-1}$) are also simulated in south-western Mediterranean.

The wind time-series at M2 (36.4912°N, 6.9611°W, in the Gulf of Cadix, west of Gibraltar Strait) in Fig. 10b shows the good agreement of the OA simulation in this area.

Figure 9c shows that at the end of the simulation (after 6 days), OA still performs well when compared to in-situ observations, for coastal as offshore locations, even if, again, there are no observations where OA simulates its highest wind values. This can also be seen in the latest days in Figure 10a,b.

The Taylor diagram in Figure 10c summarised the OA skill scores for the 7 day-period, when compared to all in-situ wind observations together, and to M1 and M2 separately. The mean bias is 1.3 m s$^{-1}$, the standard deviation is 4.1 m s$^{-1}$, and the

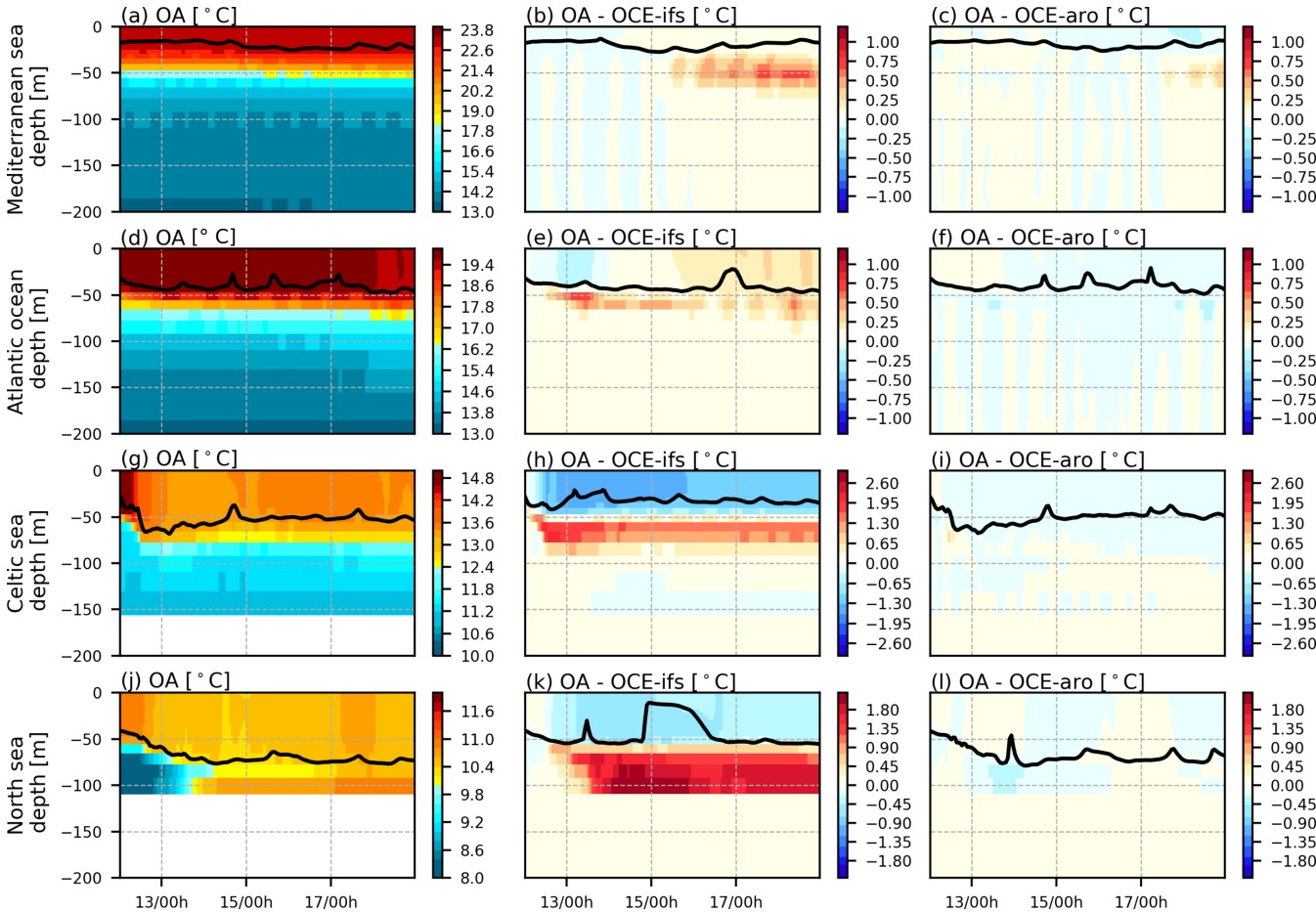

**Figure 7.** Temporal evolution of the mean vertical temperature profiles in the four zones (see Fig. 3d) simulated by the coupled (OA) simulation (a,d,g,j) and differences with the two forced ocean simulations (OA-OCE-ifs in b,e,h,k and OA-OCE-aro in c,f,i,l). The black lines delimit the averaged MLD of OA (a,d,g,j), OCE-ifs (b,e,h,k) and OCE-aro (c,f,i,l).

correlation is 0.36 on average. This bias on AROME wind speed was already identified in Rainaud et al. (2016) and Léger et al. (2016), in particular for strong wind situation and when comparing to coastal observing platforms. Further investigation would be needed to understand the origin of such systematic bias, looking into both the AROME physics and the method to diagnose
the wind at 10 meters, but is out of the scope of this paper.

### 4.1.4   Rainfall

In the OA coupled simulation, the accumulated precipitation during the 7 simulated days is shown in Figure 11a. The rain is heterogeneously distributed over the domain. In the Bay of Biscay, it follows the trajectory of Callum with rainfall reaching 200 mm in the two first simulated days (Fig. 11c). In the Aude department (Fig. 11e), where the heavy precipitating event

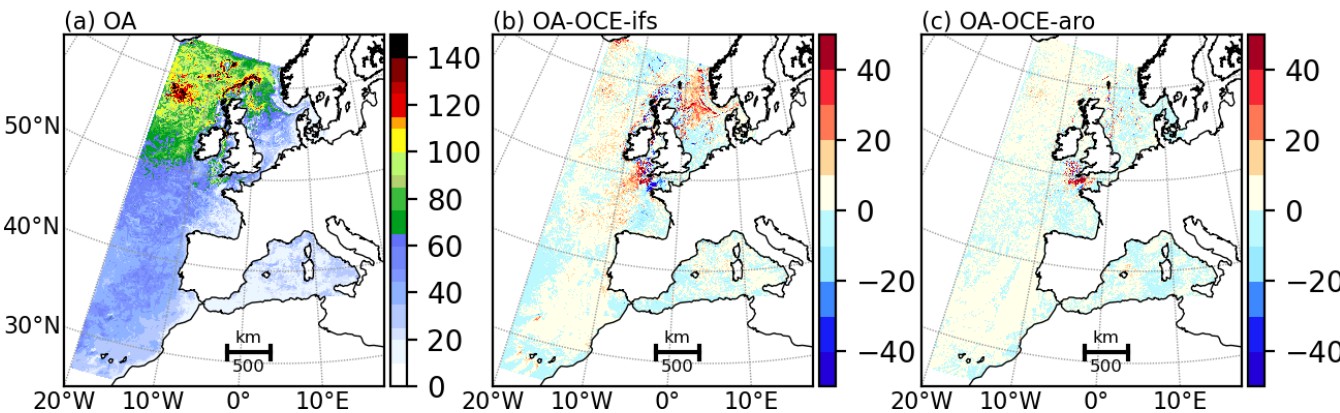

**Figure 8.** Daily-averaged oceanic mixed layer depth simulated by OA simulation on the last day of simulation (a) and its difference with OCE-ifs and OCE-aro forced simulations (b, c).

355 described in section 3.1 occurred, the simulated accumulated precipitation reaches 300 mm in 1 day as observed, but are located about 100 km to the east of the observed one. This location corresponds to the Massif Central relief (also known as the Cévennes), suggesting that the rapid and moist marine low-level flow is well reproduced, but with a slightly different orientation than observed and thus with a dominant triggering factor related to orographic uplift [whereas it was in fact related to convergence between the south-easterly flow with a cold front (Caumont et al., 2021)]. However, it is important to note that

360 the Mediterranean HPE correspond to forecast ranges between +66h and +90h for AROME, *i.e.* quite far from the standard AROME forecast operational ranges. Despite the fact that observed and simulated intense precipitation amounts are not located exactly at the same place, the heavy precipitation signature with large values of rainfall amounts in only few hours in the OA forecast, appears very valuable in the context of very early warning of such severe events.

### 4.2 Impact of OA coupling on the oceanic forecast

365 In this section, we compare NEMO forced simulations (OCE-ifs and OCE-aro) and AROME/NEMO coupled (OA) simulations (Table 2) in order to quantify the effect of OA coupling on the oceanic forecast.

#### 4.2.1 Sea surface temperature, salinity, height and currents

The effect of coupling on the temporal evolution of the oceanic surface field forecast is presented in Fig. 6. First, we can note that AROME simulates stronger winds than IFS (Fig. 6q,r,s,t), which leads to more intense oceanic vertical mixing in the OA

370 and OCE-aro simulations than in the OCE-ifs one (see next section). This effect is clearly visible along the Callum trajectory, in the Celtic or North Sea. Changing the atmospheric forcing of NEMO between IFS and AROME drastically modifies the oceanic response, with a more intense sea surface cooling for simulations using AROME (see OA in blue and OCE-aro in green in Fig. 6c,d). Thus, the effect of changing the atmospheric model to force NEMO is larger than the effect of interactive


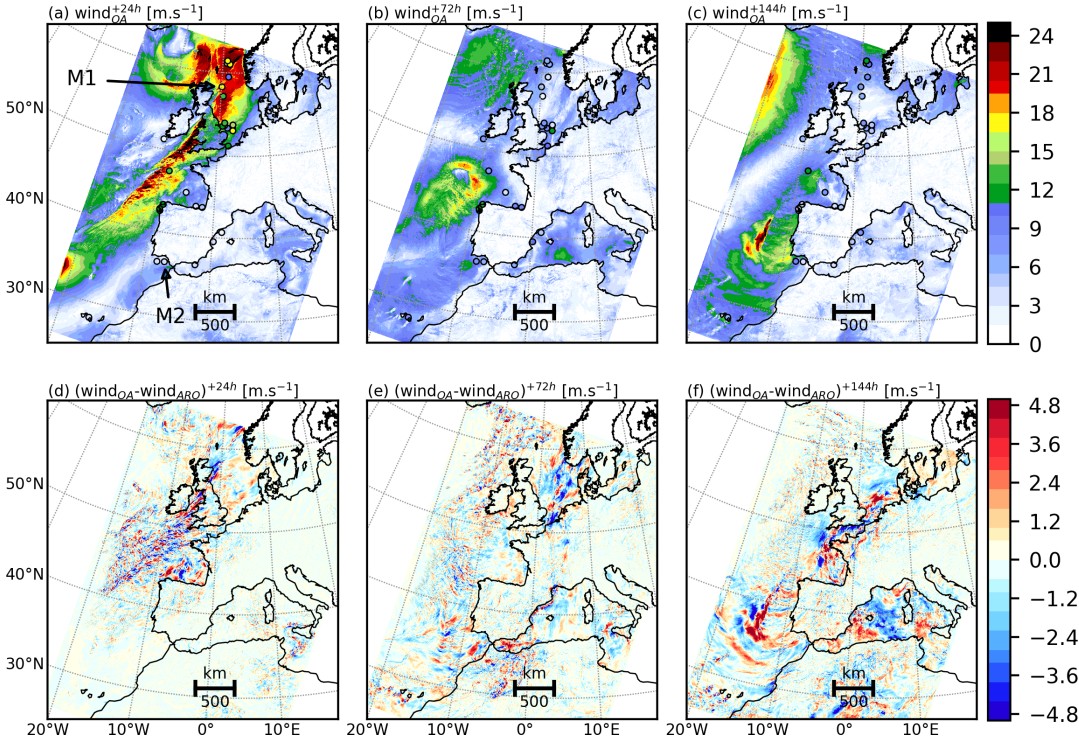

**Figure 9.** Instantaneous 10 m-ASL wind speed (m s$^{-1}$) simulated by OA (a,c,e) and differences with ARO (b,d,f) for forecast ranges of (a,b) +24h (13 Oct. 2018, 00UTC), (c,d) +72h (15 Oct. 2018 00UTC) and (e,f) +144h (18 Oct. 2018 00UTC). In (a, b, c), the colour circles represent the wind speed measured by mooring buoys at that time ; M1 and M2 labels in (a) indicate the location of the two mooring buoys used in Figure 10.

coupling on the simulated surface fields, in particular for SST and SSS forecast. However, the effect of the ocean-atmosphere

coupling on the SST and SSS induces also a feedback, leading to a more important cooling of the surface waters in coupled (OA) than in forced (OCE-aro) simulations. This sea surface cooling enhancement with coupling is in fact related to a lower non-solar net heat flux in OA (not shown), meaning a larger heat loss at night (and a lower diurnal heating) for ocean in OA than in OCE-aro. In fact, the surface cooling rapidly change the atmospheric low-level environment and stability [without significant difference in the wind speed (and wind stress)]. In particular, the coupled simulation represents an amplification

loop, as the 2m-specific humidity is progressively lower in OA (than in OCE-aro/ARO). This enhances evaporation, and thus amplify slightly the surface cooling. We can note that this effect of ocean-atmosphere coupling is visible for all boxes after 3 simulated days and differences increase until the end of the simulation (see Fig. 6a,b,c,d).

Figure 6m,n,o,p display the impact of atmospheric forcing on the sea surface currents, which are on average less intense in the OCE-ifs simulation than in the OA and OCE-aro simulations, which is explained by weaker winds in IFS than in AROME

(Fig. 6q,s,t), except for the Atlantic box where IFS wind is larger than in AROME (Fig. 6r), but no significant change in SSC


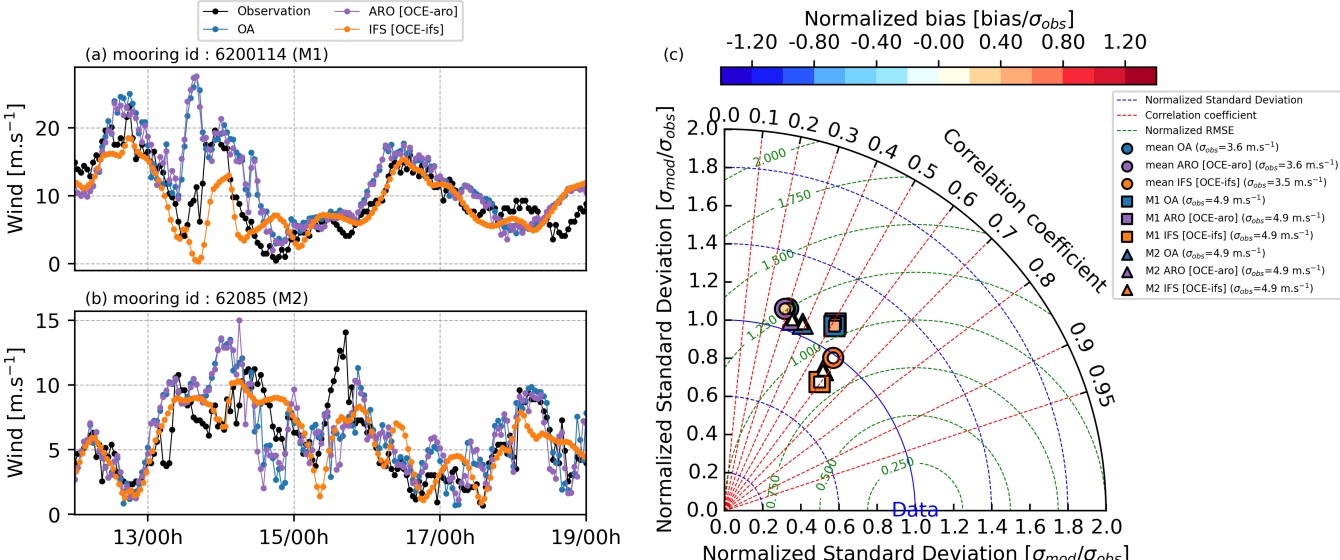

**Figure 10.** Temporal evolution of 10-m wind speed observed and simulated at the location of two moorings M1 (a) and M2 (b) (see Figure 9a for locations). (c) Taylor diagrams made for the whole dataset of 44 selected moorings in circles, for moorings M1 and M2 only in squares and triangles, respectively. The external colour indicates the experiment: blue for OA, purple for ARO [OCE-aro] and orange for IFS [OCE-ifs].

are found (Fig. 6n). Also, for the Mediterranean box, on 14-15 October, the wind is larger in IFS forecast, and the SSC is stronger in OCE-ifs during that period (Fig. 6m,q).

Regarding the SSH (Fig. 6i,j,k,l), the main signal is due to tidal oscillations. The differences between the 3 simulations are relatively small or even indistinguishable, meaning that the effect of the choice of the atmospheric forcing model or OA
coupling on SSH is of an order of magnitude smaller than the tidal forcing.

### 4.2.2 Temperature vertical profiles

The temporal evolution of the temperature profiles simulated by the coupled and forced simulations are computed for the same four boxes as previously (see Fig. 3) and compared on Figure 7 to examine the evolution of the oceanic mixed layer. The profiles located in the Mediterranean Sea (first line in Fig. 7) or in the Atlantic Ocean (second line in Fig. 7) boxes show quite
small evolution of the mixed layer during the simulations, only a cooling about 0.2°C for the Atlantic ocean box. At these locations, differences between the simulations are also quite small (Fig. 7b,c,e,f) or only related to differences in the mixing, mainly due to the wind forcing (Fig. 7b,e). In the northern part of the domain, temperature profiles show significant changes during the 7-day period. These changes are not only located in the near-surface waters (where it exceeds -2°C), but also deeper, and even below the mixed layer depth (black line in Fig. 7g,j). For the Celtic Sea and North Sea boxes, differences of the OA
simulation with the OCE-ifs simulation are large (±2.5°C corresponding to a mixing-induced dipole with cooling near the



**Figure 11.** Accumulated precipitation (mm) simulated by the coupled (OA) simulation [left column] and differences with the ARO forced simulation [right column]: (a,b) Total amounts over the 7 day-period, 24h-accumulated amounts (c,d) over British Islands between 12 Oct. 00UTC and 13 Oct. 00UTC (+00 to +24h forecast ranges), and (e,f) over Western Mediterranean area between 14 Oct. 18UTC and 15 Oct. 18UTC (between +66h and +90h forecast ranges).





surface and warming near the thermocline, Fig. 7h,k) and much higher than the differences between the OA and the OCE-aro simulations (Fig. 7i,l). More generally for the four boxes, differences are larger when comparing OCE-ifs to OA than when comparing OA and OCE-aro. This illustrates that the effect of changing the atmospheric forcing has a larger effect on ocean surface and also vertical profiles, than changing from a forced to a coupled simulation.

### 4.2.3 Oceanic boundary layer depth

Differences between ocean mixed layer depth (MLD) simulated by the three simulations (OA, OCE-ifs and OCE-aro) are represented in Figure 8. Maximum differences between OA and OCE-ifs are localised around the British Islands and can reach ± 50 meters. Here again, differences between OA and OCE-aro are smaller, even if located in the same areas (Fig. 8c) . When computing the relative differences between OA and OCE-ifs (blue bars in Fig. 12), they exceed more than 50% in the Celtic sea and 30% in the North and Mediterranean seas, while, in the Atlantic box, differences are smaller (below 5%). Computing the same MLD differences for the pairs OA vs OCE-aro (orange bars) and OCE-aro vs OCE-ifs (green bars) highlights that differences in the MLD are maximum for OA vs OCE-ifs and of the same order of magnitude between OCE-aro and OCE-ifs. As discussed in the previous Section, it means that the effect of the change in atmospheric forcing is responsible of the main signature in changes in the near-surface oceanic structure, and that the effect of the coupling only accentuates this oceanic response.

### 4.3 Impact of OA coupling on the atmospheric forecast

In this section, we compare AROME forced (ARO) and AROME/NEMO coupled (OA) simulations (Table 2), in order to quantify the impact of OA coupling on the atmospheric forecast. In the ARO simulation, the sea surface temperature (SST) is persistent and equal to the SST field used as initial condition in the OA simulation (Fig. 3a) and the oceanic surface currents are null, while in the OA simulation, the evolution of sea surface temperature and currents are taken into account.

### 4.3.1 Wind

Whether after 1, 3 or 6 days of simulation, the wind simulated by the forced (ARO) and the coupled (OA) simulations shows differences (Fig. 9d,e,f). The size of the wind difference structures appears to increase with time, as OA and ARO diverge from each other. After 1 simulated day (Fig. 9d), the maximum differences between the OA and ARO simulations are located along the Callum storm passage, where strong winds are present (Fig. 9a). They reach $\pm 5$ m.s$^{-1}$ locally, corresponding to more than 20% of the simulated 10-m wind speed. Elsewhere in the domain, effect of coupling on the 10-m wind speed is relatively small ($< 1$ m.s$^{-1}$). This suggests that, for these short-forecast ranges, coupling only changes the internal dynamics of the storm with embedded convection. After 3 simulated days (Fig. 9e), the maximum differences are now located in the western half of the domain, where the storms Callum, Leslie and Mickael have moved. They reach $\pm 4$ m.s$^{-1}$ locally and correspond to more than 100% at some locations, meaning that the low-level dynamics started to significantly diverge between the two simulations.


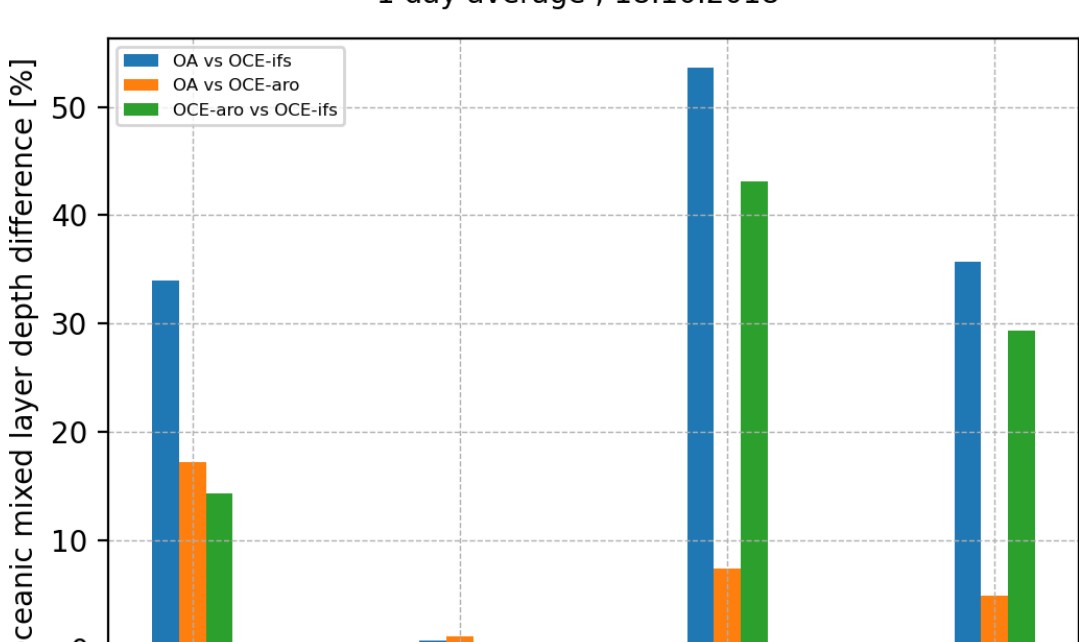

**Figure 12.** Instantaneous oceanic mixed layer depth differences between pairs of simulation after 168 simulated hours extracted in the four areas presented in Figure 3d.

After 6 simulated days (Fig. 9f), significant differences are present throughout the entire domain with quite large structures related to differences in the large-scale dynamics.

Despite these overall differences, the effect of the OA coupling does not significantly change the temporal evolution of the 10-m wind speed forecasts in comparison to OCE-aro forced simulation and to mooring data (Fig. 6 ; Fig. 10). Note that the
10-m wind speed simulated by OCE-ifs has better scores than OA and ARO simulations at the mooring locations (Fig. 10), which can be explained by the wind overestimation in OA and ARO than in the IFS forecast (as seen for M1 and M2 examples).

#### 4.3.2 Rainfall

We highlight here the impact of the OA coupling on the rainfall amounts during the 7 days, as shown in the Figure 11b. The mean accumulated precipitation over the whole domain differs between the coupled and forced simulations by less than 0.5%.
However, total rainfall amounts can vary locally by more than 100 %, especially in the north of Balearic Islands (5°E, 40°N) or close to Sicily (15°E, 38°N). Concerning the heavy precipitation that took place in Wales (Fig. 2), the differences between the OA and ARO simulations in total rainfall amounts during the first 48 hours presented in the Figure 11d are quite small. The maximum differences reach about 20 mm and represent locally up to only 10 % of the 48h-cumulated rainfall amount. These





differences are related to small displacements of the rain bands, linked to changes in the wind maxima localisation discussed
in the previous section (Fig. 9d). The effect of coupling is clearly visible for the Mediterranean heavy precipitating event
(cf. observed case in Fig. 2). Fig. 9f shows that the 24h-rainfall amounts forecast in the OA simulation diverge from the ARO
simulation. The precipitation areas are shifted in the OA simulation, which can be explained by the differences in low level wind
convergence position that is a key triggering factor for mesoscale convective systems that generate heavy precipitations. This
high sensitivity of wind convergence to sea surface structures and their evolution over north-western part of the Mediterranean
Sea was already highlighted in previous studies (e.g. Rainaud et al., 2017; Meroni et al., 2018) and is there confirmed.

## 5    Conclusions

A new forecast-oriented high-resolution ocean-atmosphere coupled system using state-of-the-art AROME (cy43) and NEMO
(3.6) models has been described and evaluated trough comparisons with observations in this paper. A new domain over Western
Europe, including the two domains used for high resolution atmospheric and oceanic forecasts at Météo-France and Mercator
Ocean International (MOI) respectively, has been designed. This coupled system was evaluated through 7-day simulations
performed around an October 2018 study case. This case was chosen because during these 7 days, three storms and two inten-
sively raining periods occur over the simulated domain, which makes it a good candidate to study ocean-atmosphere coupling
impacts, as air-sea interactions are exacerbated by such extreme conditions.

This new coupled system successfully simulates the different storms and their associated strong wind and surface turbu-
lent fluxes. The maximum precipitation values of the two extreme rainfall events are also well simulated. Oceanic response
associated with these extreme conditions shows significant vertical oceanic mixing along the storms tracks. This mixing is
responsible of an intense sea surface cooling of more than 1.5°C in some places. Comparisons with observations (satellites
and drifting buoys) show that this cooling is well localised even if too intense, notably in the Celtic Sea. This coupled system
also successfully simulates the oceanic tides with their associated sea surface height and currents variations. For this latter
parameter notably, additional investigations will be needed to further explore the role of the current-feedback implementation
in the AROME-NEMO coupled system.

To evaluate the effect of OA coupling in the atmospheric and oceanic forecast, three additional simulations have been per-
formed in a forced mode. Two simulations close to the current operational forecast systems operated at Météo-France and MOI
respectively were run, and a third simulation with NEMO was set to understand the source of the main differences for ocean
forecast. Indeed, compared to the closest simulation of the current operational system operated at MOI, the OA coupled system
has two main differences: it uses a different atmospheric model (AROME versus IFS) with higher horizontal resolution (2.5
km compared to 9 km) and represents ocean-atmosphere feedbacks. The different simulations show that the effect of changing
the atmospheric model (and in particular its associated horizontal resolution) has a greater effect on the ocean forecast than
taking into account the ocean-atmosphere feedbacks. The combined effect of both is visible on the surface fields, SST, SSS



and currents, but also on the structure of the oceanic mixed layer. It is explained by a stronger wind in the atmospheric forcing with AROME at 2.5 km horizontal resolution (+20% in some places), which leads to stronger surface fluxes, and thus to a stronger oceanic response. Sea surface cooling can be higher than 6°C in some places for our study case, it can affect the entire oceanic mixed layer, and is exacerbated where storms are located. The effect of ocean-atmosphere coupling on atmospheric forecast has been examined through comparison of simulated 10-m wind speed and accumulated precipitations with the forced simulation, in which SST is kept persistent. Modifications due to coupling appear from the first simulated hours and increase over simulated time. The SST evolution in the OA simulation leads to changes in the location of the oceanic frontal structures notably, which induce changes in the wind convergence, and thus in the location of the atmospheric convection areas and heavy rainfall. The coupling impact on the simulated wind and precipitation can vary up to 100% in some places.

In summary, the coupled system slightly changes the atmospheric forecast on average even if strong differences are found locally for 10-m wind speed and rainfall amounts, and significantly improves the sea surface temperature forecast (with a bias reduction of 30 %), when compared with the equivalent uncoupled forecast systems of Météo-France and MOI, respectively, and with the observations available over the simulation period and in our study area.

Even if other case studies are necessary, this work already highlights the relevance of high-resolution ocean-atmosphere coupling for the two kinds of forecast. It also shows the affordability of such numerical prediction system regarding the computation costs (see appendix A2) that can be shared and especially through the development of common tools. Still, future challenges remain for an operational implementation of such high-resolution coupled system, in particular the insertion of a coupled data assimilation scheme, with also the issue of the data availability for both components, and a coordinated code management with objectives of continuously improving the computing efficiency. Nevertheless, thanks to our joint work for its update, with the development and application to a new region, the AROME-NEMO coupled system permits now to further apprehend operational ocean-atmosphere coupling in both institutes, Météo-France and Mercator Ocean International.

*Code and data availability.* NEMO is available at https://www.nemo-ocean.eu/ after a user registration on the NEMO website. The version used is NEMO_v3.6. OASIS3-MCT was used in version OASIS3-MCT_4.0. It can be downloaded at https://portal.enes.org/oasis. The public may copy, distribute, use, prepare derivative works and publicly display OASIS3-MCT under the terms of the Lesser GNU General Public License (LGPL) as published by the Free Software Foundation, provided that this notice and any statement of authorship are reproduced on all copies. SURFEX open-source version (Open-SURFEX) including the interface with OASIS from v8_0 is available at http://www.umr-cnrm.fr/surfex/ using a CECILL-C Licence (a French equivalent of the L-GPL licence; http://www.cecill.info/licences/Licence_CeCILL-C_V1-en.txt), but with exception of the gaussian grid projection, the LFI and FA I/O formats, and the dr HOOK tool. Although the operational AROME code cannot be obtained, the modified sources for cy43 are available on demand to the authors for the partners of the ACCORD consortium and are included in the new Météo-France official release based on cycle 48 (cy48t1). Outputs from all simulations discussed here are available upon request to the authors.



The moored and drifting buoys data were collected and made freely available by the Coriolis project and programmes that contribute
to it (http://www.coriolis.eu.org). FES2014 was produced by Noveltis, Legos and CLS and distributed by Aviso+, with support from Cnes
(https://www.aviso.altimetry.fr/).

*Author contributions.*  All authors (JP, JB, CLB, GS, GF and GG) contributed to the conceptualisation and methodology of the study as well
as drafting, reviewing and editing the article. GF finalized the Vortex/Olive-Swapp experimental configuration for coupled simulations and
extracted the IFS forecast files. The configurations NEMO-eNEATL36 and AROME-Mercator were developed by JP and JB, who also ran
the coupled and uncoupled simulations. JP, JB, CLB, GS and GG carried out the analysis of the results.

*Competing interests.*  The authors declare that they have no conflict of interest.

*Acknowledgements.*  This work was funded by Mercator Ocean International. The authors thank Sylvie Malardel, Soline Bielli (LACy),
Sébastien Riette (CNRM) and the SWAPP system team (Météo-France) who helped us in the implementation of the coupled experiment
design in the Vortex/Olive-Swapp environment.




## Appendix A: Technical information

### A1    Coupling masks between NEMO and AROME

Figure A1 presents the masked parts of each domain. The black areas in Figure A1a correspond to where NEMO does not resolve the ocean. In AROME (Fig. A1b), the masked area corresponds to the same unsolved areas of NEMO plus the northern, western and southern extensions. There, in the OA coupled experiment, AROME sees null current and keeps the initial SST field constant in time.

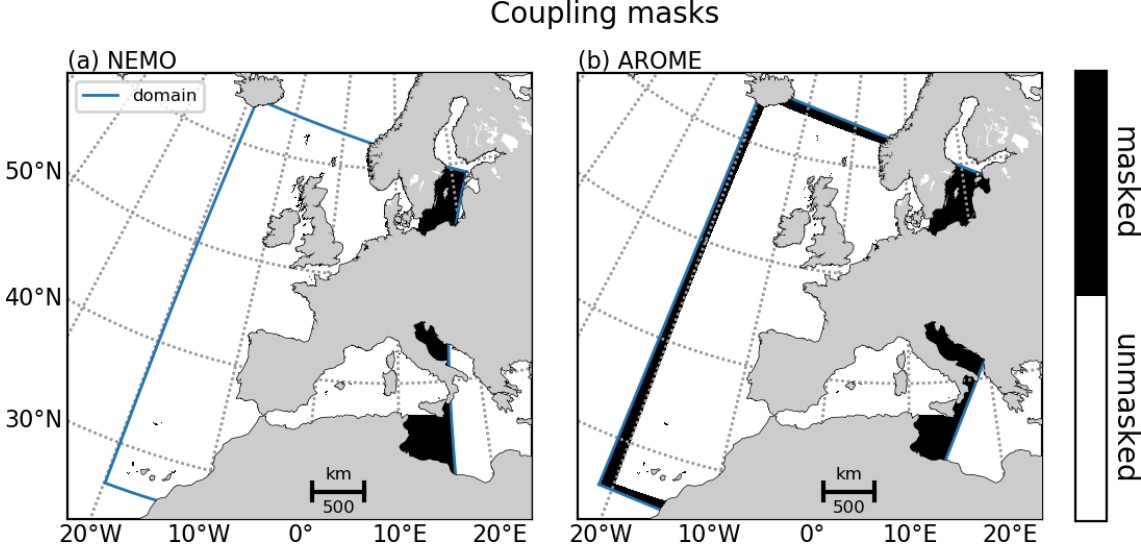

**Figure A1.** Coupling masks between NEMO and AROME.

### A2    Simulation environment and High Performance Computing characteristics

All the developments are performed using Vortex/Olive python-based framework, used to run AROME operational simulations at Météo-France. This coupling system is running on the new Météo-France supercomputer belenos (https://www.top500.org/system/179853/). In total, this supercomputer has 294 912 cores on 2 307 nodes and a peak performance of approximately 10.5 PFlop/s. Each nodes have a Random Access Memory (RAM) of 256 GB minimum.

Table A1 summarises the computational cost of the different simulations presented in this article (Tab. 2).

The coupled simulation runs on 15 nodes and 424 cores corresponding to 12 nodes and 384 cores for AROME, 2 nodes and 32 cores for NEMO and 1 node and 8 cores for XIOS. Simulated time is roughly 12 h for AROME (ARO) and AROME/NEMO (OA) simulations indicating that the effect of OASIS coupler is negligible for this coupled system. The OA simulation CPU cost does not exactly correspond to the sum of the executions of AROME and NEMO/XIOS, as NEMO cores pass some time to wait AROME fields in this configuration. It is indeed superior to the 18 432 CPU hours for one AROME forced (ARO)





simulation plus the CPU cost of the oceanic model and the XIOS server for coupled AROME/NEMO (OA) simulation and finally corresponds to a 20 % total additional CPU cost (23 040 CPU hours). Note that simulated time of NEMO simulations alone (OCE-aro and OCE-ifs simulations) are roughly equal to 8.5 h (with 2 nodes and 32 cores for NEMO and 1 node and 8 cores for XIOS) corresponding to CPU cost of approximately 3 280 CPU hours (14.2 % of the CPU cost of the OA coupled system). For the purpose of this comparison, we used the same number of nodes for NEMO simulations alone (OCE-aro and OCE-ifs simulations) as the one used in AROME/NEMO simulations but it can be optimised, for example, by increasing the number of used cores by node.

**Table A1.** Elapsed time and computational cost of the different 7-day simulations. 1 node contains 128 cores and CPU cost is equal to elapsed time by the number of nodes by 128 (the number of cores by nodes) whatever the true number of nodes effectively used.

| Simulation | Elapsed time | Nb nodes | CPU cost |
|---|---|---|---|
| OA | $\approx 12$ h | 15 | 23 040 h |
| ARO | $\approx 12$ h | 12 | 18 432 h (80% of OA) |
| OCE-ifs / OCE-aro | $\approx 8.5$ h | 3 | 3 280 h (14 % of OA) |





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
