# Peer review of "Development of a forecast-oriented km-resolution ocean-atmosphere coupled system for Western Europe and sensitivity study for a severe weather situation"

_Natural Hazards and Earth System Sciences, 2021_

## Referee Comment (RC1)

**Review of "Development of a forecast-oriented km-resolution ocean-atmosphere coupled system for Western Europe and evaluation for a severe weather situation" by Joris Pianezze et al., https://doi.org/10.5194/nhess-2021-226**

This study by Pianezze et al. presents a kilometer-scale atmosphere-ocean coupled system newly developed to improve the forecasts over the northeastern Atlantic and western Mediterranean seas. Additionally, during the 12-19 October 2018 storm event, on one hand, the performances of the new system are assessed and, on the other hand, a sensitivity study of the impact of the coupling is performed.

I believe the development and use of km-scale atmosphere-ocean models should be promoted as it has been proven in many studies that these models improve both forecasts and climate projections. However, in my opinion, this particular study has failed to demonstrate the interest of such numerically expansive modelling suite and thus cannot be published without taking into account the major corrections described below.

**Major comments:**

(1) Design of the coupled system:

At the strategic level, it is extremely difficult to understand why the modelling system does not use similar grids in the atmosphere and ocean and thus reduce the computations by exchanging the fields between the two grids without any interpolation. In the actual configuration the size of the two grids is nearly identical and the fields are interpolated, so there is absolutely no computational gain and, in my opinion, no justification for such a strategy.

Additionally, the design of the ocean model which ignores the Baltic Sea, the Adriatic Sea and the border east of Sicily is a bit strange as it covers seas that in fact are not modelled and then could have been limited to the previous operational setup (NEATL36). Overall, it feels that atmospheric and ocean grids were designed separately and patch up together for the coupled system.

This leads to my last point. Not imposing SST from ocean model to the atmospheric model particularly on the open Atlantic boundary seems a strange choice. It will definitely create a discontinuity in the SST field imposed on the atmospheric model and avoiding these kind of discontinuities which can translate into numerical "shocks" is a basic modelling concept. In brief, I would strongly recommend to rethink a modelling strategy that fluidly allows the atmospheric and ocean models to exchange fields over their entire domains without any grey zones (as presented in Fig. 1 and A1). At the very least, the authors must merge figures 1 and A1 and discuss at length the different drawbacks of their modelling strategy and how they could be fixed and why they are not.

(2) Evaluation of the modelling system:

First, testing the model on one storm event does not represent de facto an evaluation of a complex atmosphere-ocean modelling suite. It is merely a test of the capacity/performance of the model during a specific event. However, the authors did a nice sensitivity study on

the impact of the coupled system on the results. I would thus recommend to highlight the sensitivity study aspect and not the evaluation aspect in their article. A more appropriate title for their study could be something like: "Sensitivity to coupling of a forecast-oriented km-resolution ocean-atmosphere system during a severe weather situation".

Second, the authors mention the configuration of the operational system actually in use and composed of AROME (AROME-France, 1.3 km-resolution) and NEMO over the Iberia-Biscay-Ireland (IBI) region (NEATL36, 1/36°-resolution). The comparison of the performances of this system with their newly developed system thus seems a mandatory step to show the interest of their developments. This is an important missing part of the study and some clues on how the new model outperforms (or not) the already widely use system should be provided.

Third, the sensitivity to coupling should be presented for all comparison with observations. For example, Fig. 4 should show the spatial differences also for $SST_{OCE-ifs}$-$SST_{sat}$ & $SST_{OCE-aro}$-$SST_{sat}$.

Fourth, the structure of section 4 is extremely hard to follow and overall confusing. In my opinion, for each sub-section, the authors should systematically, first present the performance against observations (for all experiments) and then the comparison between experiments. (e.g. 4. Sensitivity to coupling, 4.1 Sea surface temperature, 4.1.1 performance, 4.1.2 sensitivity, 4.2 Temperature, salinity, height, currents and ocean mixed layer, 4.2.1 performance, 4.2.2 sensitivity, 4.3 wind, 4.3.1 performance, etc.).

Finally, some sub-sections do not have any comparison with observations (e.g. 4.1.2 or 4.1.4). If finding observations in the ocean is a difficult task and should be acknowledge, I believe that the authors can have access to many land-based coastal weather stations that would provide observations of rain but also wind. I would also recommend to check the availability of Argo float measurements or other ocean observations (e.g. CTD) during the time of the numerical experiments. Overall, this study could benefit from a bigger number of observations to assess the performances of the different experiments. Indeed, for the moment, some comparisons done in Fig. 6, Fig. 7, Fig. 8, Fig. 11 and Fig. 12 seem relatively pointless as we truly don't know how each experiment performs against observations for any the compared variables.

(3) Conclusions of the study:

From the presented results, it is clearly shown that IFS performs better than AROME (coupled or uncoupled) (i.e. better comparison for wind speed measurements and comparison with SST buoys similar for all ocean experiments) and thus (again with the limited presented comparisons with observations) it feels pointless to develop a complex atmosphere-ocean model which will use a lot of numerical resources while a NEMO model forced with IFS seems to provide better results. I truly believe that the presented results do not properly reflect the performances of the newly developed model but as it stands the study does not demonstrate the interest of such a model and this should be emphasis in the conclusions and in the abstract.

**Specific comments:**

The following list of specific comments is pretty succinct as major comments should first be addressed before a more detailed review can be made.

Lines 60-64: Efforts done by diverse research groups in the world to develop km-scale atmosphere-ocean coupled operational forecast models should be mentioned (e.g. Indian and western Pacific Oceans, Hawaii, UK, Adriatic Sea, etc.)

Lines 165: feedback of ocean currents to atmospheric models is not state-of-the-art coupling and may be developed a bit more here.

Figure 1. Please combine with Figure A1 for discussion of the modelling strategy concerning the grids (see major comments).

Figure 2. Scale of the different sub-plot makes it confusing to understand firsthand what the different subplot are representing. Plotting coastline and drawing sub-domains (b, c, d) on top of satellite image (panel a) may help reader identify without effort the geographical locations of interest.

Figure 3. Not sure how this figure is relevant concerning the sensitivity study done in the article. It is obviously expected that during a storm event SST will diverge from initial (i.e. background) conditions. I think this figure is not needed in the main article and may be presented as supplementary material to highlight how the area of interest correspond to the obtained results.

Figure 4. Please include comparison of satellite data with all experiments (see major comments).

Figure 5. The legend on the top of the time series as well as on the right hand side of the figure should be increased in size.

Figure 6. Please remove wind from the plot and put wind comparison in a separate figure dealing with wind comparison specifically.

Figures 7 & 8. Interesting figure only if all experiments presented are previously compared with measurements over the full "water column" (e.g. Argo floats, CTDs) in the different zone presented. Otherwise, no conclusion can be drawn except this experiment is similar or different from the coupled one.

Figure 9. Please include comparison $wind_{OA}$ - $wind_{IFS}$. Labels on the top of figures should also be increased in size.

Figure 10. As Figure 2.

Figure 11. Please include $rain_{OA}$ - $rain_{IFS}$

Figure 12. As for Figures 7 & 8, interesting only if all experiments are previously compared to observations in the different zone presented.

Data availability: From my understanding of the EGU publication rules (I may be wrong), the specific model results (atmosphere and ocean) used in this study should be publicly available as

anyone should be able to reproduce the presented findings of the article, but authors mentioned it is only "upon request".

---

## Author Comment (AC1)

Reply to Reviewers

manuscript number : **nhess-2021-226**
entitled : *"Development of a forecast-oriented km-resolution ocean-atmosphere coupled system for Western Europe and evaluation for a severe weather situation"*
authors : Joris Pianezze, Jonathan Beuvier, Cindy Lebeaupin Brossier, Guillaume Samson, Ghislain Faure, and Gilles Garric
* * *
**Reviewer 1**
* * *
**Review of "Development of a forecast-oriented km-resolution ocean-atmosphere coupled system for Western Europe and evaluation for a severe weather situation" by Joris Pianezze et al., https ://doi.org/10.5194/nhess-2021-226**

This study by Pianezze et al. presents a kilometer-scale atmosphere-ocean coupled system newly developed to improve the forecasts over the northeastern Atlantic and western Mediterranean seas. Additionally, during the 12-19 October 2018 storm event, on one hand, the performances of the new system are assessed and, on the other hand, a sensitivity study of the impact of the coupling is performed. I believe the development and use of km-scale atmosphere-ocean models should be promoted as it has been proven in many studies that these models improve both forecasts and climate projections. However, in my opinion, this particular study has failed to demonstrate the interest of such numerically expansive modelling suite and thus cannot be published without taking into account the major corrections described below.

Thank you for your constructive review of our article. You will find our answers to your comments (in blue) below, with clarifications for changes done in the manuscript.

**Major comments :**

(1) Design of the coupled system :

At the strategic level, it is extremely difficult to understand why the modelling system does not use similar grids in the atmosphere and ocean and thus reduce the computations by exchanging the fields

between the two grids without any interpolation. In the actual configuration the size of the two grids is nearly identical and the fields are interpolated, so there is absolutely no computational gain and, in my opinion, no justification for such a strategy.

Additionally, the design of the ocean model which ignores the Baltic Sea, the Adriatic Sea and the border east of Sicily is a bit strange as it covers seas that in fact are not modelled and then could have been limited to the previous operational setup (NEATL36). Overall, it feels that atmospheric and ocean grids were designed separately and patch up together for the coupled system.

This leads to my last point. Not imposing SST from ocean model to the atmospheric model particularly on the open Atlantic boundary seems a strange choice. It will definitely create a discontinuity in the SST field imposed on the atmospheric model and avoiding these kind of discontinuities which can translate into numerical "shocks" is a basic modelling concept. In brief, I would strongly recommend to rethink a modelling strategy that fluidly allows the atmospheric and ocean models to exchange fields over their entire domains without any grey zones (as presented in Fig. 1 and A1). At the very least, the authors must merge figures 1 and A1 and discuss at length the different drawbacks of their modelling strategy and how they could be fixed and why they are not.

The starting point of our coupled system development is the NEATL36/IBI operational system. It inherits of the MOI global system ORCA grid which is a tri-polar grid projection. The NEATL36 domain is a regional ORCA grid at 1/36°-resolution, where the Baltic, Adriatic and Tyrrhenian Seas are not represented. In the Mediterranean Sea, the eastern open boundaries of NEATL36 are located along a line across Corsica and Sardinia.

For operational oceanography purposes, a relatively small extension of this domain was decided, notably motivated by the objective to better cover the whole continental french coasts. The NEATL36 has thus been extended in eNEATL36 in order to cover the whole Western Mediterranean Sea (and so the eastern coasts of the French Riviera and Corsica). So, the insertion of the Tyrrhenian Sea is the only change of the basin we considered, and consequently eNEATL36, as NEATL36, does not reproduce the Baltic and Adriatic Seas. Looking at Figure 1, one can see that in fact the Adriatic Sea is disconnected from the rest of our ocean domain, but information comes across the Mediterranean open boundary which is now a southern boundary located across the Sicily Channel. The Baltic Sea is of course fully included in the global system, that is used to provide ocean lateral conditions at the eastern open boundary between Sweden and Poland.

The AROME domain was then after chosen to cover the eNEATL36 domain, but knowing that the ORCA projection is not an option for AROME (the development of such grid projection would have been a huge effort with an unknown gain). Knowing that identical grids were not reachable in our case, we choose to have a larger atmospheric domain than for ocean. This choice is quite common for regional coupled

systems, and is argued by the fact that i) SST is well observed/analysed and thus a SST field completion appears easier and ii) considering air-sea interactions, a SST discontinuity seen by the atmosphere could be considered as less critical than a wind (or wind stress) discontinuity seen by the ocean. The AROME domain has been rotated (compared to the AROME-France domain) in order to have parallel western boundaries for AROME and NEMO, but also we took care to have a small band to avoid spurious fields related to the Davies' zones of AROME to affect NEMO.

Thus with this grids configuration, the SST seen by AROME in OA indeed combines the explicit NEMO SST with the initial analysed SST close to its boundaries. Figure A shows the full SST field seen by AROME at the initial time and after 7 days of simulation. Indeed, as in the grey zones of AROME the

[Figure]

FIGURE A – Sea surface temperature (°C) seen by AROME (mask and unmasked areas) in the OA experiment at the initial time (20181012 0000UTC, left panel) and at the end of the simulation (20181019 0000UTC, right panel).

[Figure]

FIGURE B – 2 m-specific humidity (left, g/kg), 2 m-air temperature (middle, °C), and 10 m-wind speed velocity (right, m/s) at the end of the OA simulation (20181019 0000UTC).

SST comes from the 1/12°-resolution global system of MOi, and also stays constant during the coupled forecast, a more marked SST discontinuity appears progressively during the forecast. The largest SST discontinuity appears close to the western boundary, because of the marked ocean surface cooling in OA (see Figure 3d in the manuscript). SST discontinuities in the Sicily Channel or at the exit of the Baltic Sea are less discernible. To evaluate the effects of these discontinuities on the simulated 2m air humidity, temperature and 10 m wind speed in the OA simulation, the AROME simulated fields are presented in Fig. B for the +168h forecast-range, *i.e.* after 7 days of forecast when the SST discontinuities are the most significant. We can note that only a relatively small signature of the SST discontinuities is found at low level, notably in the southern (south-western) part of the domain, while elsewhere (in the Sicily Chanel or in the Baltic Sea for instance), the atmospheric near surface parameters are not affected by the SST

[Figure]

FIGURE *C* – **New Figure 1 (Old Figure 1 merged with old Figure A.1). Caption :** *Simulation domain illustrated by the bathymetry [m] in NEMO (in blue) and by the orography [m] of the AROME model (in green-brown colors). The lines indicate the boundaries of NEMO-eNEATL36 configuration (red) and of the AROME-Mercator domain (black); the green lines highlight the open boundaries in the oceanic model. For AROME-Mercator, the grey and orange marine zones are always uncoupled (constant initial SST and null current are used, see text). For eNEATL36, the orange marine zones are not solved in the regional oceanic simulations. The dashed lines indicate the boundaries of the actual operational configurations of AROME (AROME-France, 1.3 km-resolution, in black) and NEMO over the Iberia-Biscay-Ireland (IBI) region (NEATL36, 1/36°-resolution, in red).*

discontinuity (Fig. B). In particular, the northern and western boundaries of AROME are in fact more largely constrained by the incoming atmospheric lateral conditions and their space-time variability.

To manage the two different grids, the interpolation of the exchanged fields is done by OASIS through the remapping option, distributed on the different computing process units, with weight fields computed in advance. The numerical cost of interpolations during the forecast is completely negligible. Note that the same kind of interpolations is necessary in the forced mode.

To better explain the simulation strategy and the grids/masks management, the Figure presenting the simulation domain has been reviewed as suggested by merging Figure 1 with Figure A.1 (see Figure C, here). For clarity, the open boundaries of the new NEMO configuration have been highlighted with green lines. Several text modifications have been inserted at the beginning of section 2 and in section 2.3 about the coupling strategy. Also, sections 2.1 and 2.2 have been reversed so the reader could better follow the domains design (ocean first then atmospheric model).

(2) Evaluation of the modelling system :

First, testing the model on one storm event does not represent de facto an evaluation of a complex atmosphere-ocean modelling suite. It is merely a test of the capacity/performance of the model during a specific event. However, the authors did a nice sensitivity study on the impact of the coupled system on the results. I would thus recommend to highlight the sensitivity study aspect and not the evaluation aspect in their article. A more appropriate title for their study could be something like : "Sensitivity to coupling of a forecast-oriented km-resolution ocean-atmosphere system during a severe weather situation".

We agree that, as we only investigate one case, the term "sensitivity" appears more appropriate than "evaluation" that implies a larger period of verification with successive forecasts, but also a careful verification of improvements compared to current operational runs. The study presented here is more a sensitivity study to coupling done with two forecast-oriented models.

The new title of the paper is : "*Development of a forecast-oriented km-resolution ocean-atmosphere coupled system for Western Europe and **sensitivity study** for a severe weather situation*"

Second, the authors mention the configuration of the operational system actually in use and composed of AROME (AROME-France, 1.3 km-resolution) and NEMO over the Iberia-Biscay-Ireland (IBI) region (NEATL36, 1/36°-resolution). The comparison of the performances of this system with their newly developed system thus seems a mandatory step to show the interest of their developments. This is an important missing part of the study and some clues on how the new model outperforms (or not) the already widely

use system should be provided.

The actual operational systems are uncoupled and are operated separately by Météo-France for Numerical Weather Prediction and by Mercator Ocean International/Puerto del Estados for regional operational oceanography over the IBI area.

At Météo-France, AROME is used for various high-resolution regional forecast kinds. Over continental France, AROME is used :

— four times a day (00, 06, 12 and 18 UTC) for 42 or 48h-range deterministic forecasts at 1.3 km-resolution (see the domain in Figure 1, dashed black box), using a 3D-var assimilation scheme and the ARPEGE forecasts at its boundaries. This is called AROME-France ;

— twice a day (00 and 12 UTC) for 48h-range deterministic forecasts at 2.5 km-resolution (same domain extension as AROME-France), using IFS forecasts as atmospheric initial and boundary conditions, and AROME-France analysis for surface initial conditions including SST. No assimilation is done. This forecast set-up is called AROME-IFS in the paper and hereafter.

— four times a day (03, 09, 15, 21 UTC) for 45h-range ensemble predictions with 16 members (12 members before July 2019) at a 2.5 km-resolution (same as AROME-IFS). It is called PEARO forecasts.

The closest operational configuration of AROME we can compare to our coupled model is thus AROME-IFS, but knowing that in October 2018, AROME-IFS used a former code version (cy42, against cy43t2 in our study) and used initial conditions of IFS at a coarser resolutions (both horizontal and vertical) to drive AROME than our study. It uses the ARPEGE SST analysis (against the global [GLO12] MOi analyses in this study) and it covers a smaller domain (the same as AROME-France, see Fig. 1) and a shorter period (until 00 UTC 14 October, for the forecast starting on 12 October 00 UTC). This means than comparing ARO with AROME-IFS necessitate to consider all these features, while the ARO/OA comparison clearly shows the interactive coupling impact. In fact, the AROME-IFS operational forecast starting on 12 October 2018, 00UTC, shows results quite close to the IFS forecast. As an example, the wind forecast by AROME-IFS is presented in Figure D and compared to the OA experiment (see also Fig. J d,g), as well as to satellite observations. The wind induced by Callum in AROME-IFS appears clearly more realistic. We still need to investigate several hypothesis, notably in the AROME physical options, to explain the Callum wind intensification overestimation in our ARO and OA experiments. Nevertheless, the comparison of ARO with OA stays fully valid to evaluate the sensitivity to coupling.

For operational oceanography, the regional system IBI (Irish Biscay Iberia area) is operated weekly for the Copernicus Marine Service. It is based on the NEATL36 model configuration. The current version of this system was not in operation in October 2018, thus only the sequence of analyses is available as operational products for that period of time (no forecasts). To stay close to the version of the IBI/NEATL36 system currently in operation, we chose to perform the OCE-ifs simulation, which we consider as a re-

[Figure]

FIGURE *D – Comparison of the 10 m-wind speed forecast for 12 October 2018 1200 UTC by (a) OA and (b) the operational AROME-IFS forecast (starting on 12 October 2018 00 UTC) with (c) wind satellite observations (ASCAT) for 12 October 2018, at 11 08UTC.*

forecast of the 12-19 October 2018 period by a system close to the actual operational one, except for the small spatial extension in the Tyrrhenian Sea.

In summary, we chose to insert the ocean-atmosphere coupling in the most recent code versions of the two regional operational systems. The experimental set-up was designed to investigate, during a sequence of severe events, the high-resolution atmospheric representation and the coupling impacts on both forecasts, with the clear objective to avoid other numerical changes that can perturb the comparison. We agree again that the "evaluation" term in the initial title was not accurate and would have necessitate the full comparison with operational real-time runs. We think the new title referring to a "sensitivity study" is now more relevant.

Third, the sensitivity to coupling should be presented for all comparison with observations. For example, Fig. 4 should show the spatial differences also for $SST_{OCE-ifs}$-$SST_{sat}$ & $SST_{OCE-aro}$-$SST_{sat}$.

Following your comment, Figure 4 has been modified to include a comparison with simulations OCE-ifs, OCE-aro and ARO (Fig. E). We also remove the simulated SST fields and only plot the differences between the simulated SST and the satellite observation at the end of the simulation (+168 h).

Fourth, the structure of section 4 is extremely hard to follow and overall confusing. In my opinion, for each sub-section, the authors should systematically, first present the performance against observations (for all experiments) and then the comparison between experiments. (e.g. 4. Sensitivity to coupling, 4.1 Sea surface temperature, 4.1.1 performance, 4.1.2 sensitivity, 4.2 Temperature, salinity, height, currents and

[Figure]

FIGURE E – **New Figure 4. Caption :** *Comparison with L3 satellite SST observations at the end of the simulation (19 October 2018 00 UTC) : differences (in ° C) with (a) ARO SST, (b) OA simulated SST, (c) OCE-aro simulated SST and (d) OCE-ifs simulated SST.*

ocean mixed layer, 4.2.1 performance, 4.2.2 sensitivity, 4.3 wind, 4.3.1 performance, etc.).

We are sorry the initial structure of the results presentation, with a subsection dedicated to the coupled simulation examination, was difficult to follow with these go-backs in the figures.

The results section has been fully reviewed and re-ordered following your comment, which was also

raised by the second reviewer. We chose to keep two subsections : "4.1. Oceanic forecast 4.1.1. Sea surface temperature 4.1.2. Sea surface dynamics, salinity and ocean mixed layer" and "4.2. Atmospheric forecast 4.2.1. Wind 4.2.2. Rainfall".

Finally, some sub-sections do not have any comparison with observations (e.g. 4.1.2 or 4.1.4). If finding observations in the ocean is a difficult task and should be acknowledge, I believe that the authors can have access to many land-based coastal weather stations that would provide observations of rain but also wind. I would also recommend to check the availability of Argo float measurements or other ocean observations (e.g. CTD) during the time of the numerical experiments. Overall, this study could benefit from a bigger number of observations to assess the performances of the different experiments. Indeed, for the moment, some comparisons done in Fig. 6, Fig. 7, Fig. 8, Fig. 11 and Fig. 12 seem relatively pointless as we truly don't know how each experiment performs against observations for any the compared variables.

For rainfall (former section 4.1.4), the IFS forcing does not cover land surfaces. This is due to the operational extraction done at MOi, e.g. before interpolation land values are masked and oceanic values are extrapolated onto land domain to avoid taking land values into account when interpolating over the ocean. Rainfall estimation over sea are also too rare to perform any validation. The conclusion of our sensitivity study is that ARO and OA produce quite similar forecast in term of rainfall chronology and large-scale structures, but with some convective systems displacements that can lead to large differences locally (see Figure 13 (old Figure 11) b,f over the Balearic Sea for instance).

For salinity (former section 4.1.2), an operational validation tool was applied on our two forced ocean-only experiments (OCE-ifs and OCE-aro simulations). Figure F(c,d) presents the comparison to 13.5 m-depth salinity in-situ observations through Root Mean Square Error for the OCE-ifs and OCE-aro simulations. It shows that the change of atmospheric forcing for AROME improves very slightly the salinity RMSE over the period and the whole domain [and that coupling impact is neutral (OA results are similar to OCE-aro, not shown)], but the robustness of this result is difficult to estimate due to the scarcity of in-situ salinity observations.

Applied for temperature at the same depth (Fig. F a,b), the validation tool confirms that better scores are obtained on average for OCE-ifs than for OCE-aro, but local improvements are found when using AROME forcing, notably along Cornwall coasts in the Celtic Sea. Figure F(a,b) has been added in the paper (new Fig. 8) with comments on the simulations representation of the ocean mixed layer that rely on these scores.

(3) Conclusions of the study :

From the presented results, it is clearly shown that IFS performs better than AROME (coupled or

[Figure]

FIGURE *F – Comparison with in-situ profiling platforms (Argo floats, CTD profiles, mooring, gliders and drifting buoys) for (a,b) temperature in $^{circ}C$ and (c,d) salinity in psu, at 13.5 m-depth. (a,c) Differences in RMSE between OCE-ifs and OCE-aro (blue means better scores for OCE-ifs). (b,d) RMSE time-series for OCE-aro and OCE-ifs experiments.* **Subplots a and b are now inserted in the manuscript (New Figure 8).**

uncoupled) (i.e. better comparison for wind speed measurements and comparison with SST buoys similar

for all ocean experiments) and thus (again with the limited presented comparisons with observations) it feels pointless to develop a complex atmosphere-ocean model which will use a lot of numerical resources while a NEMO model forced with IFS seems to provide better results. I truly believe that the presented results do not properly reflect the performances of the newly developed model but as it stands the study does not demonstrate the interest of such a model and this should be emphasis in the conclusions and in the abstract.

Despite the wind overestimation associated to Callum in ARO/OA, the AROME performance compares quite well with IFS. And thus considering the whole simulated period and domain, OCE-ifs and OCE-aro appears quite similar except the too large cooling in the Celtic Sea. The new Figure F which compares OCE-aro and OCE-ifs scores for temperature and salinity at 13.5m, tends also to moderate the conclusion of a better performance of IFS. AROME represents accurately the storms trajectories, and is able to produce a heavy precipitation event over South-Eastern France with a good timing despite a long forecast range. The atmospheric processes represented by AROME have a finer scale ; it is know that such fine scale representation can lead to a "double penalty" phenomena (e.g Rossa et al., 2008; Crocker et al., 2020) when compared to punctual observations, especially with a scattered observation network as for atmospheric in-situ measurement over sea. This double-penalty effect can also affect the ocean compartment by combination of its own high-resolution and of the atmospheric forcing/coupling high resolution, with for instance local salinity response to high-resolution precipitation.

As you pointed out, this study is a first sensitivity analysis based on one case study only. To fully conclude on the coupling benefit, a more systematic comparison must be done. From the NWP point of view, the numerical cost of coupling AROME to NEMO is estimated to be around +5%. Even if the ratio benefits / costs must be more carefully examined, considering the low additional cost and the number of AROME operational forecast instances run daily, we confidently think that high-resolution coupled forecasts are reachable for 2025-2030. We agree that the coupling appears costly for operational oceanography. However, we think we have introduced the scientific and technical interests to commonly build this coupled numerical tool and demonstrated, in that sense, the coupling affordability for mutual applications towards operations.

**Specific comments :**

The following list of specific comments is pretty succinct as major comments should first be addressed before a more detailed review can be made.

— Lines 60-64 : Efforts done by diverse research groups in the world to develop km-scale atmosphere-ocean coupled operational forecast models should be mentioned (e.g. Indian and western Pacific

Oceans, Hawaii, UK, Adriatic Sea, etc.)

This part of the introduction (lines 60-64) was written more to highlight the additional effort implied by the objective of running operational forecasts with coupled systems, based on the synthesis papers of Brassington et al. (2015) and Pullen et al. (2017) that gather the results of several research groups. It appears difficult to give an exhaustive view of the diverse research groups targeting operational coupled systems. However, new references have been added in the paragraph before, to mention notably the coupled forecast model development efforts for operational purposes over the Maritime Continent (Thompson et al., 2021), over the Gulf of St-Lawrence (Pellerin et al., 2004) or also in the Adriatic Sea (Ličer et al., 2016; Vilibić et al., 2018).

— Lines 165 : feedback of ocean currents to atmospheric models is not state-of-the-art coupling and may be developed a bit more here.

As shown in Renault et al. (2019), the current feedback, by causing surface stress anomalies, modulates the oceanic circulation by slowing down the mean oceanic circulation and dampening the mesoscale activity. To take into account the current feedback, it is necessary to use the relative winds in the computation of air–sea fluxes, i.e., the difference between the near-surface winds and the surface oceanic currents, instead of absolute winds. Because of the implicit treatment of the bottom boundary condition in most atmospheric models, the use of relative winds also necessitates a modification of the tridiagonal problem associated with the discretization of the vertical turbulent viscosity. This second modification was not initially include in the AROME-NEMO coupling of Rainaud et al. (2017) and Lebeaupin Brossier et al. (2017).

The following sentence has been added in Section 2.2 (old 2.1) of the paper to explain a bit more this current feedback implementation : "*The surface current acts in two ways on turbulence by using the relative winds, i.e., the difference between the near-surface winds and the surface oceanic currents, instead of absolute winds (i) in the computation of air–sea fluxes and (ii) in the tri-diagonal problem associated with the discretization of the vertical turbulent viscosity because of the implicit treatment of the bottom boundary condition in the atmospheric model. Only the first effect was included in the former AROME-NEMO couplings (Rainaud et al., 2017; Lebeaupin Brossier et al., 2017; Sauvage et al., 2021).*"

— Figure 1. Please combine with Figure A1 for discussion of the modelling strategy concerning the grids (see major comments).

As you suggested, Figure 1 has been reviewed to combine the previous Figure 1 and A.1 in order to highlight the masked and unresolved areas and to discuss them in the "Coupling strategy" section. This new figure is presented in Fig. C. See also our answer to your major comment.

— Figure 2. Scale of the different sub-plot makes it confusing to understand firsthand what the different

subplot are representing. Plotting coastline and drawing sub-domains (b, c, d) on top of satellite image (panel a) may help reader identify without effort the geographical locations of interest.

To better identify the locations of subplots (b,c,d) on the map with a wide geographic coverage (a), we added colored contours to the different subplots and reported them on the larger one. We also added a dashed outline showing the area of interest. The new figure is presented in Fig. G.

[Figure]

FIGURE *G* – ***New Figure 2 of the paper. Caption :*** *Illustrations of the case study : (a) True color image of Terra/MODIS (source : https ://worldview.earthdata.nasa.gov/) on 11 October 2018 over the North Atlantic Ocean showing the storm Callum and the Leslie and Michael hurricanes (arrows depict their trajectories towards the area of interest); (b) Rainfall totals (mm) from 11 to 12 October 2018 over Wales (Callum's impacts, Figure 64 from Kendon et al., 2019, source: MetOffice); (c) Wind gust observations (km/h) over Iberian Peninsula on 13 October 2018 around 23 UTC (Leslie's landfall, source : www.meteociel.fr); (d) Rainfall amounts (mm) between 06 UTC on 14 October and 06 UTC on 15 October 2018 over the French Languedoc region (Aude event, source : Météo-France - edited 19/02/2019).*

— Figure 3. Not sure how this figure is relevant concerning the sensitivity study done in the article. It is obviously expected that during a storm event SST will diverge from initial (i.e. background) conditions. I think this figure is not needed in the main article and may be presented as supplementary material to highlight how the area of interest correspond to the obtained results.

Since Figure 4 now shows sea surface temperature differences, we kept Figure 3 in the core of the article, which is the only figure that shows a SST map. It serves also as a quantification of the SST modifications not represented in ARO.

— Figure 4. Please include comparison of satellite data with all experiments (see major comments).
Done. See also our answer to the major comments.

— Figure 5. The legend on the top of the time series as well as on the right hand side of the figure should be increased in size.

The size of the legends on the top of the time series as well as on the right hand side one has been increased.

— Figure 6. Please remove wind from the plot and put wind comparison in a separate figure dealing with wind comparison specifically.

Done. We have removed the wind comparison in Figure 6 (the new Figure 6 is presented in Figure H) and we present now the temporal evolution of the wind speed and the rainfall in a new figure (Fig. I).

— Figures 7 & 8. Interesting figure only if all experiments presented are previously compared with measurements over the full "water column" (e.g. Argo floats, CTDs) in the different zone presented. Otherwise, no conclusion can be drawn except this experiment is similar or different from the coupled one.

The purpose of these Figures is to present the sensitivity of vertical profiles between all the simulations and thus to show the sensitivity to forcing / coupling in the four areas of interests. The areas of interests were identified according to the coupled model results only (*i.e.* without any consideration of the presence of ocean profile observations).

— Figure 9. Please include comparison wind OA - wind IFS. Labels on the top of figures should also be increased in size.

Done. Wind from the IFS forcing has been added in Figure 11 (old Figure 9), but removed from the difference to include buoys measurements. The new figure 11 is presented in Figure J.

[Figure]

FIGURE *H* – **New Figure 6 of the paper. Caption :** *Temporal evolution of simulated sea surface temperature (SST, °C), salinity (SSS, psu), height (SSH, m) and current speed (SSC, m s$^{-1}$) extracted in the four areas presented in Figure 3d. Note that ARO does not have SSS nor SSH.*

— Figure 10. As Figure 2.

This Figure has been changed as Figure 5.

— Figure 11. Please include rainOA - rain IFS

The IFS forcing fields used to drive OCE-ifs is at a lower resolution than the native resolution and in addition only include rainfall forecasts (as other atmospheric parameters) on the sea points (as usually done by the MOi operational download and treatment procedure). We cannot therefore fairly compare the OA simulation with IFS.

— Figure 12. As for Figures 7 & 8, interesting only if all experiments are previously compared to observations in the different zone presented.

The purpose of this Figure is to present the coupling sensitivity and to better quantify the ocean mixed layer responses due to the high-resolution forcing on one side from the response due to the interactive cou-

[Figure]

FIGURE *I* – **New Figure 13 of the paper. Caption :** *Temporal evolution of simulated 10m wind speed (m.s⁻¹ ; a,b,c,d) and rainfall (mm/h ; e,f,g,h) extracted in the four areas presented in Figure 3d.*

pling on the other side.

— Data availability : From my understanding of the EGU publication rules (I may be wrong), the specific model results (atmosphere and ocean) used in this study should be publicly available as anyone should be able to reproduce the presented findings of the article, but authors mentioned it is only "upon request".

The section "Statement on the availability of underlying data" of NHESS journal data policy page (https ://www.natural-hazards-and-earth-system-sciences.net/policies/data_policy.html) indicates that "If the data are not publicly accessible, a detailed explanation of why this is the case is required.". The coupled and atmosphere-only simulations are not reproducible, due to the AROME code licence which is not public. However, the simulation results are publicly available, but represent a huge volume of data. Consequently our preference goes to the extraction of the simulation results "upon request", as seen in some other NHESS papers, and as indicated here in the Data availability statement.

[Figure]

FIGURE *J* – **New Figure 11 of the paper. Caption :** *Instantaneous 10 m-ASL wind speed (m s⁻¹) simulated by OA (a,b,c), ARO (d,e,f) and IFS seen by OCE-ifs (g,h,i), for forecast ranges of (a,d,g) +24h (13 Oct. 2018, 00UTC), (b,e,h) +72h (15 Oct. 2018 00UTC) and (c,f,i) +144h (18 Oct. 2018 00UTC). The colour circles represent the wind speed measured by mooring buoys at that time ; M1 and M2 labels in (a) indicate the location of the two mooring buoys used in Figure 12.*

**Reviewer 2**

— I sugget to homogenize the way you present the references to used upstream data. For example, introducing ECMWF IFS instead of simply IFS (ln. 123) or global-IFS (which I guess it is the same). IBI is also cited as IBI36 : better to use one unique reference.

Thanks for your comment. We homogenize the references to IFS and IBI. We chose to keep IFS when we refer to ECMWF IFS global operational forecast and IBI when we refer to IBI configuration at 1/36°.

— the user may not know what IBI is. I see you added correct references, however since your model is focusing on a different implementation (since the spatial domain is wider) I suggest to remove in Section 2.2 the specificiation of where the IBI boundary was or simply specify why you are proposing this new spatial domain more clearly.

Done. We added definition if IBI in the text and not only in the caption of Figure 2.

— you refer to CMEMS many times. It would be good to introduce it once, at the beginning, and cleaning the paper from redundant references like in Section 4.1.1 ln. 261-262.

We added the definition of CMEMS the first time we use it and clean some redundant references.

— It is clear you decided to split presentation of results between validation and evaluation of impact. The impression I got is that in the evaluation of the OA model, the explanation can be a bit confused since you present also forced experiments in the plots : in fact, in the second part of the section where you present the impact of OA coupling, you use the previous plots and explain it. I would focus on discussing evaluation of the OA coupling system directly when you describe the impact, so something like this :
  — 4.1 Validation and evaluation of OA coupling on ocean forecast highlighting if and how OA improves skills wrt forced system
  — 4.2. Impact of OA coupling on the atmospheric forecast

Following your comment, also common with the first reviewer, the results section has been fully reviewed and re-ordered. We have decided to consider only 2 sections and 2 sub-sections per section : "4.1. Oceanic forecast 4.1.1. Sea surface temperature 4.1.2. Sea surface dynamics, salinity and ocean mixed layer" and "4.2. Atmospheric forecast 4.2.1. Wind 4.2.2. Rainfall".

— ln. 141-143 : could you please specify the dataset/reference you used for the river runoff ?

Thanks for your comment. We add that information and modify the text in consequences : *"As in the operational IBI configuration (Sotillo et al., 2015, 2021), rivers freshwater inputs are imposed part as daily OBC in the domain locations for 33 main rivers and part as a climatological coastal runoff to close the water budget from land. For the 33 main rivers explicitly considered, flow-rate data are based on a combination of daily observations, simulated data (from SMHI E-HYPE hydrological model) and climatology (monthly climatological data from GRDC and French "Banque Hydro" dataset)."*

— ln. 236 : which bulk are you using in the OCE-ifs ? It would be nice to specify them.

The sea surface bulk parametrization used in OCE-ifs simulation is the IFS parametrization described in the ECMWF documentation (ECMWF, 2020) and available in the SBC code section of NEMO. This reference was also added in the paper.

— ln. 261-263, the SST L3 you are using is missing the reference : could you add it ? I think you are using SST daily at night-time and not SST daily average observation : is it ?

SST L3 product we use comes also from the CMEMS website (http ://marine.copernicus.eu). It corresponds to the multisensor merged, $0.02°$ and daily average product. The explicit product code and reference document (Orain et al., 2021) have been added in the article.

[revised manuscript text omitted]

---

## Author Response (AR1)

**Reply to Editor**

**manuscript number: n hess-2021-226**

entitled : "Development of a forecast-oriented km-resolution ocean-atmosphere coupled system for Western Europe and evaluation for a severe weather situation"

new title : "Development of a forecast-oriented km-resolution ocean-atmosphere coupled system for Western Europe and sensitivity study for a severe weather situation"

authors : Joris Pianezze, Jonathan Beuvier, Cindy Lebeaupin Brossier, Guillaume Samson, Ghislain Faure, and Gilles Garric

Editor decision : Reconsider after major revisions (further review by editor and referees) - 4 Jan 2022 by Piero Lionello

**Comments to the author :**

Dear Dr. Pianezze and Beuvier :

Based on the reviewers' evaluations and your replies in the interactive discussion, your manuscript will be reconsidered after a major revision (further review by editor and referees). You are requested to submit a revised version of your manuscript, where your text is modified according to the reviewers' suggestions and the interactive discussion. The final publication of your manuscript in NHESS is conditional upon your answers to the reviewers and the changes in the revised version of your manuscript. Please, submit together with the revised version, a copy of your manuscript where all your changes are annotated and the list of your detailed answers to the reviewers' comments.

I am particularly concerned by your admission that this study does not represent a validation of the high resolution coupled system, but a sensitivity study, because only one case study is considered (therefore the statistics is quite limited). The sensitivity of the forecast to the coupling and to the resolution is well known in the literature and it would not justify a publication in a high-level journal. An interesting study should explain whether there is an improvement with respect to the reference uncoupled forecast. It seems that you have already some encouraging results of the SST forecast but none on the weather forecast. However, you explain that the effect of changing the atmospheric model is much larger than that of the coupling. This also somehow reduces the relevance of your results for supporting the use of this coupled model system.

Looking forward to receiving your revised manuscript, the replies to these comments and to those of the reviewers Best regards Piero Lionello Dear Dr. Lionello,

Thanks for your comments.

Following yours and the reviewers' comments, we have totally revised the result section of the first submitted version of the paper to focus on the coupling sensitivity (rather than the evaluation of the coupled system which as it was raised, cannot be done robustly on a single case study). We have also added some comparisons with observations and improved some figures. As it was suggested, we have also discussed in more depth the differences between the coupled simulation and the forced simulations that are close to the actual operational systems.

Indeed, our results are in the same line than previous sensitivity studies related to coupling and horizontal resolution. The different oceanic simulations show that the largest effect is brought by the change of the atmospheric model (and its associated horizontal resolution) and the coupled system slightly changes the oceanic and atmospheric forecasts on average, but with strong differences found locally. Even if other case studies are necessary, we think this work shows scientific and technical advances, for the two forecast kinds, especially (i) the relevance of building high-resolution ocean-atmosphere coupling for operational purposes, (ii) the affordability of the AROME-NEMO system regarding the computation costs and (iii) the benefits of common development tools with shared expertise.

You will find in the following our responses to the two anonymous reviewers' comments, as provided in the interactive discussion, and the submission of the revised version of the manuscript in the MS overview.

**Sincerely,**

Joris Pianezze and Jonathan Beuvier, on behalf of all the co-authors

Reply to Reviewers

manuscript number : nhess-2021-226

entitled : "Development of a forecast-oriented km-resolution ocean-atmosphere coupled system for Western Europe and evaluation for a severe weather situation"

authors : Joris Pianezze, Jonathan Beuvier, Cindy Lebeaupin Brossier, Guillaume Samson, Ghislain Faure, and Gilles Garric

Reviewer 1

Review of "Development of a forecast-oriented km-resolution ocean-atmosphere coupled system for Western Europe and evaluation for a severe weather situation" by Joris Pianezze et al., https://doi.org/10.5194/nhess-2021-226

This study by Pianezze et al. presents a kilometer-scale atmosphere-ocean coupled system newly developed to improve the forecasts over the northeastern Atlantic and western Mediterranean seas. Additionally, during the 12-19 October 2018 storm event, on one hand, the performances of the new system are assessed and, on the other hand, a sensitivity study of the impact of the coupling is performed. I believe the development and use of km-scale atmosphere-ocean models should be promoted as it has been proven in many studies that these models improve both forecasts and climate projections. However, in my opinion, this particular study has failed to demonstrate the interest of such numerically expansive modelling suite and thus cannot be published without taking into account the major corrections described below.

Thank you for your constructive review of our article. You will find our answers to your comments (in blue) below, with clarifications for changes done in the manuscript.

**Major comments :**

(1) Design of the coupled system :

At the strategic level, it is extremely difficult to understand why the modelling system does not use similar grids in the atmosphere and ocean and thus reduce the computations by exchanging the fields between the two grids without any interpolation. In the actual configuration the size of the two grids is nearly identical and the fields are interpolated, so there is absolutely no computational gain and, in my opinion, no justification for such a strategy.

Additionally, the design of the ocean model which ignores the Baltic Sea, the Adriatic Sea and the border east of Sicily is a bit strange as it covers seas that in fact are not modelled and then could have been limited to the previous operational setup (NEATL36). Overall, it feels that atmospheric and ocean grids were designed separately and patch up together for the coupled system.

This leads to my last point. Not imposing SST from ocean model to the atmospheric model particularly on the open Atlantic boundary seems a strange choice. It will definitely create a discontinuity in the SST field imposed on the atmospheric model and avoiding these kind of discontinuities which can translate into numerical "shocks" is a basic modelling concept. In brief, I would strongly recommend to rethink a modelling strategy that fluidly allows the atmospheric and ocean models to exchange fields over their entire domains without any grey zones (as presented in Fig. 1 and A1). At the very least, the authors must merge figures 1 and A1 and discuss at length the different drawbacks of their modelling strategy and how they could be fixed and why they are not.

The starting point of our coupled system development is the NEATL36/IBI operational system. It inherits of the MOI global system ORCA grid which is a tri-polar grid projection. The NEATL36 domain is a regional ORCA grid at  $1/36^{\circ}$ -resolution, where the Baltic, Adriatic and Tyrrhenian Seas are not represented. In the Mediterranean Sea, the eastern open boundaries of NEATL36 are located along a line across Corsica and Sardinia.

For operational oceanography purposes, a relatively small extension of this domain was decided, notably motivated by the objective to better cover the whole continental french coasts. The NEATL36 has thus been extended in eNEATL36 in order to cover the whole Western Mediterranean Sea (and so the eastern coasts of the French Riviera and Corsica). So, the insertion of the Tyrrhenian Sea is the only change of the basin we considered, and consequently eNEATL36, as NEATL36, does not reproduce the Baltic and Adriatic Seas. Looking at Figure 1, one can see that in fact the Adriatic Sea is disconnected from the rest of our ocean domain, but information comes across the Mediterranean open boundary which is now a southern boundary located across the Sicily Channel. The Baltic Sea is of course fully included in the global system, that is used to provide ocean lateral conditions at the eastern open boundary between Sweden and Poland.

The AROME domain was then after chosen to cover the eNEATL36 domain, but knowing that the ORCA projection is not an option for AROME (the development of such grid projection would have been a huge effort with an unknown gain). Knowing that identical grids were not reachable in our case, we choose to have a larger atmospheric domain than for ocean. This choice is quite common for regional coupled

systems, and is argued by the fact that i) SST is well observed/analysed and thus a SST field completion appears easier and ii) considering air-sea interactions, a SST discontinuity seen by the atmosphere could be considered as less critical than a wind (or wind stress) discontinuity seen by the ocean. The AROME domain has been rotated (compared to the AROME-France domain) in order to have parallel western boundaries for AROME and NEMO, but also we took care to have a small band to avoid spurious fields related to the Davies' zones of AROME to affect NEMO.

Thus with this grids configuration, the SST seen by AROME in OA indeed combines the explicit NEMO SST with the initial analysed SST close to its boundaries. Figure A shows the full SST field seen by AROME at the initial time and after 7 days of simulation. Indeed, as in the grey zones of AROME the

FIGURE A – Sea surface temperature (°C) seen by AROME (mask and unmasked areas) in the OA experiment at the initial time (20181012 0000UTC, left panel) and at the end of the simulation (20181019 0000UTC, right panel).

---

## Author Response (AR2)

Reply to Editor

manuscript number : **nhess-2021-226**
new title : *"Development of a forecast-oriented km-resolution ocean-atmosphere coupled system for Western Europe and sensitivity study for a severe weather situation"*
authors : Joris Pianezze, Jonathan Beuvier, Cindy Lebeaupin Brossier, Guillaume Samson, Ghislain Faure, and Gilles Garric

*Editor decision : Reconsider after major revisions (further review by editor and referees) - 19 Feb 2022 by Piero Lionello*

*Comments to the author :*
*Dear Dr. Pianezze and Beuvier :*
*both reviewers are positive about your manuscript.*

*However, reviewer #2 is has two residual minor comments and is willing to read again your manuscript*

*Moreover, before sending your text to reviewer #2, I have the following suggestions. In case you do not agree, or I have misunderstood the text, please explain. My suggestions are :*

*line 10 : replace "persistent" with "constant"... otherwise clarify what you mean with "persistent"*

*line 11-14 I suggest to replace*
*"When compared to the operational-like ocean forecast, simulated oceanic fields show a large sensitivity to coupling. Forced ocean simulations highlight that this sensitivity is mainly controlled by the change in the atmospheric model used to drive NEMO (AROME vs. ECMWF IFS operational forecast)."*
*with*
*"Simulated oceanic fields show a large sensitivity to coupling when compared to the operational-like ocean forecast. However, forced ocean simulations highlight that this sensitivity is mainly controlled by the change in the atmospheric model used to drive NEMO (AROME vs. ECMWF IFS operational) forecast), and not by the coupling itself."*

*Line 14 The oceanic boundary layer depths can vary by more than 40%.*
*It is not clear respect to which reference. please rephrase.*

*Please ADD to the present text with track track changes also these modifications and mark them in red, without removing the previous ones. In this way all your changes with respect to the previous version will be highlighted.*

*Formally this will be treated as a major revision (because your revised manuscript will be sent to a reviewer). However, in practice, all comment s are minor in my opinion.*

*Best regards*
*Piero Lionello*

Dear Dr. Lionello,

Following yours and the reviewers' suggestions, we have reviewed the abstract, conclusion and Figure 2. We have also standardised the use of abbreviations.

You will find in the following our brief responses to the two anonymous reviewers and the revised version of the manuscript in the MS overview (the new text corrections appear in purple).

Sincerely,

Joris Pianezze and Jonathan Beuvier, on behalf of all the co-authors

Reply to Reviewers

manuscript number : **nhess-2021-226**
entitled : *"Development of a forecast-oriented km-resolution ocean-atmosphere coupled system for Western Europe and sensitivity study for a severe weather situation"*
authors : Joris Pianezze, Jonathan Beuvier, Cindy Lebeaupin Brossier, Guillaume Samson, Ghislain Faure, and Gilles Garric
* * *
**Reviewer 1 - Report #2**
* * *
**Suggestions for revision or reasons for rejection (will be published if the paper is accepted for final publication)**

I would like to acknowledge the efforts made by the authors to thoroughly address all my comments/advices/criticisms. I found the new article largely improved and only have two minor additional comments which are :

— first, despite still being septic concerning the modelling set-up used by the authors, I think only the future operational use of the modelling suite will reveal its quality and the need (or not) for coupled atmosphere-ocean kilometer-scale models in this area of the world ;

— second, I still find difficult to properly assess the strengths of the new OA model compared to the other models forced by IFS and ARO as some results only show differences without any clue on how close to "reality" are any of the models. I don't have any suggestion concerning how to achieve a better assessment and thus I think the only potential addition to the article is to slightly change the conclusions in order to reflect that maybe the presented results can't fully show the quality of the new model.

In any case, I think the article is now worthy of publication and do not need to see any updated version of the article if the authors decide to take into account my second comment.

Thank you for your review of our article.

We agree that a pre-operational use of the coupled system is mandatory to fully investigate the quality/benefits and prones/costs of ocean-atmosphere coupled kilometer-scale forecasts for both operational oceanography and numerical weather predictions.

The conclusion of the paper has been further reviewed in order to properly assess the results, and at the end to describe the main lines of the future work needed to progress towards a fair evaluation of the new coupled system and then towards any (pre-)operational implementation.
* * *
**Reviewer 2 - Report # 1**
* * *
**Suggestions for revision or reasons for rejection (will be published if the paper is accepted for final publication)**

Figure 2 : would it be possible to improve it in the readability ? In particular :
— Figure (a) has the yellow box ("b") over the white area and it is difficult to read, same box blue ("d") : maybe using a transparent effect on the background of the boxes could help.
— Figure (b) : legend is a bit small
— Figure (c) : missing legend to explain the 2D map and wind values are really small to read
— Figure (d) : legend is a bit small

Please check once more the consistency of all used abbreviations over the paper : for example, global CMEMS configuration at 1/12 is referred as GLO12 in Section 2.1 and then it becomes GLO in Section 3.2.

Thank you very much for your careful reading of the revised paper.

The set of colors for boxes in Figure 2 has been revised and the legends of maps b and d have been enlarged. The abbreviations in the paper have also been checked.